# Robustness of Mixtures of Experts to Feature Noise

Dong Sun [1]    Rahul Nittala [1]    Rebekka Burkholz [1]

## Abstract

Despite their practical success, it remains unclear why Mixture of Experts (MoE) models can outperform dense networks beyond sheer parameter scaling. We study an iso-parameter regime where inputs exhibit latent modular structure but are corrupted by feature noise, a proxy for noisy internal activations. We show that sparse expert activation acts as a noise filter: compared to a dense estimator, MoEs achieve lower generalization error under feature noise, improved robustness to perturbations, and faster convergence speed. Empirical results on synthetic data and real-world language tasks corroborate the theoretical insights, demonstrating consistent robustness and efficiency gains from sparse modular computation.

## 1. Introduction

Transformer-based models have achieved significant success in a range of deep learning tasks, including natural language processing (Brown et al., 2020; Achiam et al., 2023; Yang et al., 2024) and computer vision (Dosovitskiy et al., 2021; Liu et al., 2021). A key driver behind this progress has been the observation of "Scaling Laws"—which posit that model performance scales predictably with the number of parameters and computational budget (Kaplan et al., 2020). Consequently, the field has seen the development of ever-larger models, such as GPT-4, containing over a trillion parameters (Achiam et al., 2023). However, this rapid growth has led to a massive escalation in computational demands, training costs, and environmental impact (Strubell et al., 2020).

In response to these challenges, Mixture-of-Experts (MoE) models have emerged as a powerful architecture designed to decouple model size from computational cost (Jacobs et al., 1991; Shazeer et al., 2017). Unlike traditional "dense" models that activate all parameters for every input, MoE architec-

tures employ conditional computation (Bengio et al., 2015). They are composed of numerous smaller sub-networks, or "experts," and a gating network (router) that selects only a fraction of these experts for any given input. This paradigm allows models to scale to enormous total parameter counts while maintaining a low inference cost. A prime example of this efficiency is the Mixtral $8 \times 7$B model (Jiang et al., 2024). Despite having a total parameter count of 47B, it utilizes only about 13B active parameters per token. Remarkably, it matches or even outperforms the dense Llama-2-70B model (Touvron et al., 2023) on standard benchmarks while using significantly fewer computational resources. This phenomenon challenges the traditional view of scaling: MoEs demonstrate that sparse activation can achieve the performance of much larger dense models without the corresponding computational burden.

Despite their empirical success, the theoretical understanding of why MoEs outperform dense models remains incomplete. Existing theoretical works, such as Chen et al. (2022), largely focus on the enhanced expressiveness of MoEs. However, these analyses typically assume that each expert in the MoE is comparable in size to the dense counterpart. Under this assumption, the MoE possesses a significantly larger total parameter count, granting it a trivial capacity advantage. While this explains gains from "scaling up," it fails to isolate the inherent architectural advantages of the MoE structure itself. Furthermore, it does not fully explain scenarios where MoEs achieve superior sample efficiency or robustness when controlled for total capacity. Recent work on patch-level routing in CNNs (Chowdhury et al., 2023) has begun to address this by comparing MoEs and dense models of similar capacity, showing benefits in sample efficiency. However, their analysis is limited to an uncommon setting, the expert-choice routing, where each expert selects a fixed number of inputs.

In this paper, we identify a novel mechanism governing the success of MoEs: their ability to handle feature noise through activation sparsity, which is originally observed in ReLU-based LLMs (Zhang et al., 2022). To isolate this mechanism, we propose a theoretical framework that ensures a fair comparison: We align the total number of parameters of the MoE with that of the dense model. We model the data generation process involving a latent block-diagonal structure—representing distinct tasks or features suitable

[1]CISPA Helmholtz Center for Information Security, Saarbrücken, Germany. Correspondence to: Dong Sun <dong.sun@cispa.de>.

*Proceedings of the $43^{rd}$ International Conference on Machine Learning*, Seoul, South Korea. PMLR 306, 2026. Copyright 2026 by the author(s).

for specialized experts—observed through noisy features. In an ideal, noiseless setting, a dense model could theoretically learn this structure perfectly, rendering it equivalent to an MoE. However, in realistic scenarios where inputs are corrupted by feature noise (e.g., noisy activations inside a deep network), the dense model suffers from interference across all dimensions. In contrast, we show theoretically that the MoE's sparse activation acts as a filter, suppressing noise from irrelevant feature blocks.

Our investigation is motivated by the observation that such latent modular structures exist even within the dense activations of modern LLMs, which use variants of ReLUs as activation functions. As illustrated in Figure 1, techniques like activation pruning reveal that underlying features often exhibit block-diagonal patterns masked by noise.

The main objective of this paper is to theoretically explain how this activation sparsity exploits modularity to provide robustness against feature noise. We demonstrate that this mechanism allows MoEs to achieve superior generalization performance, training convergence speed, and sample complexity compared to dense counterparts of the same size.Beyond the theoretical characterization, we connect our framework to **linear probing** on frozen representations, where an MoE can be viewed as a set of routed linear predictors. In experiments on frozen LLM activations, MoE-based probes exhibit improved robustness to feature noise compared to global linear baselines, supporting the theory's implications for modular and sparse predictive mechanisms. Finally, our results also offer a theoretical basis for "dense-to-sparse" conversion techniques, such as MoEfication (Zhang et al., 2022) and LLaMA-MoE (Zhu et al., 2024).

**Contributions** In summary, we make the following contributions:

- We identify a novel mechanism how MoEs can outperform their dense counterpart: They can be more robust to feature noise.

- In contrast to contemporary theory on MoEs, we investigate not only generalization performance, but also highlight other benefits: enhanced robustness to certain input perturbations, faster training convergence speeds, and potentially better sample complexity.

- We instantiate our theory in the linear probing setting on frozen LLM representations by interpreting MoEs as routed linear predictors, and empirically demonstrate improved robustness to feature noise relative to global linear baselines.

## 2. Related Work

**Theoretical understanding of MoE** Early work established a universal approximation theorem for MoEs with softmax gating and linear experts (Nguyen et al., 2016). Chen et al. (2022) showed that MoEs with softmax gating and CNN experts outperform single-expert models by exploiting cluster structure in data; however, their analysis grants the MoE a substantially larger total parameter count. Chowdhury et al. (2023) moved closer to a parameter-matched regime with expert-choice routing, demonstrating sample efficiency benefits in a patch-level CNN setting. Recent theoretical contributions address a broader set of questions: task-dependent scaling, showing that MoEs improve memorization more than reasoning under fixed active-parameter budgets (Jelassi et al., 2025); expert specialization and catastrophic forgetting in continual learning (Li et al., 2025); convergence of MoE training via an EM and mirror-descent perspective (Fruytier et al., 2025); feature-learning and width-transfer guarantees via $\mu$-parameterization (Pióro et al., 2025); expressivity gains from finer expert granularity (Boix-Adsera & Rigollet, 2025) or from structured low-dimensional tasks (Wang & E, 2025); loss-landscape geometry through linear mode connectivity in MoE architectures (Tran et al., 2025); and feature-learning dynamics showing that expert recovery provably guides router learning during joint training (Liao & Kyrillidis, 2026). These directions are complementary to ours: rather than focusing on task-dependent scaling, continual learning, optimization geometry, or expressivity, we identify and analyze a distinct mechanism—*robustness to feature noise under a strict iso-parameter comparison*—that has not been studied in prior work.

**Robustness of MoE** Puigcerver et al. (2022) advanced the theoretical understanding of adversarial robustness by proving that MoEs have significantly smaller Lipschitz constants than their dense counterparts. Zhang et al. (2023) proposed a robust training method for CNN-based MoE models that uses alternating adversarial training on both the gating network and the experts, significantly improving stability against attacks. Zhang et al. (2025) incorporated robustness into MoE models while maintaining high accuracy by the introduction of a dual-model strategy, which combines standard and robust MoEs to achieve a balanced trade-off between robustness and accuracy. Li et al. (2023a) suggests that sparse MoE models are inherently strong domain-generalizable learners. By assigning distinct knowledge domains to different experts, MoE architectures can naturally handle domain shifts more effectively than monolithic dense models. This aligns with our paper's claim that the structural properties of MoE contribute fundamentally to its enhanced robustness, yet, in our case robustness with respect to feature noise.

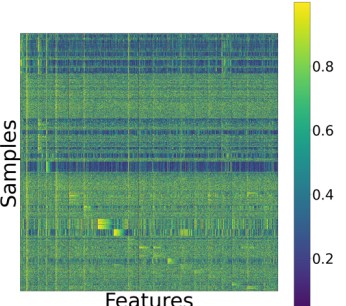 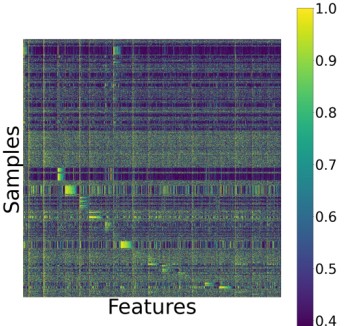 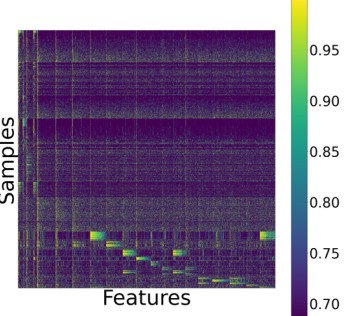

*Figure 1.* Modular structure in input activations to the `up_proj` layer within the MLP block of *layer-0* of the Llama-2-7B model, revealed using TEAL (Liu et al., 2025), for uniform magnitude pruning. From left to right, panels show activation percentiles after pruning activations based on 0%, 40%, and 70% *uniform* activation sparsity thresholds, respectively. Since the activation distributions are skewed due to the presence of outlier activations, we plot activation percentiles.

**Activation sparsity**   Our theoretical setting is motivated by work that exploits activation sparsity to convert pre-trained LLMs into MoEs. Activation sparsity, characterized by a significant portion of zero-valued entries in a model's hidden states, naturally emerges in the intermediate states of traditional ReLU-based transformers (Li et al., 2023c). Early work utilized this sparsity to accelerate LLM inference by optimizing data transfer, such as avoiding weight channel transfers to GPU registers (Liu et al., 2023) or reducing weight movement during CPU offloading (Song et al., 2024). However, modern LLMs often employ non-ReLU activations, such as SwiGLU (Shazeer, 2020) and GeGLU (Team et al., 2024). Consequently, recent research has focused on inducing activation sparsity in these newer architectures. Strategies include replacing activations like SiLU or GeLU with ReLU variants (e.g., standard ReLU or squared ReLU) combined with continued pretraining or regularization (Mirzadeh et al., 2024; Zhang et al., 2024), and developing specialized pruning methods. For instance, CATS (Lee et al., 2024) enables training-free partial sparsity in SiLU models via magnitude pruning on gate outputs, while TEAL (Liu et al., 2025) achieves 40-50 percent model-wide sparsity by pruning low-magnitude activations based on observed zero-mean unimodal distributions in LLaMA-style models.

**Error-in-variable regression / noisy features**   Our analysis of feature noise is related to error-in-variable (EIV) regression. EIV, also termed measurement error, addresses scenarios where independent variables (predictors or features) are subject to noise or measurement errors. Despite being common in practice, it presents a departure from standard regression which assumes predictors are exact (Bickel & Ritov, 1987), which is theoretically better tractable. Ignoring these errors can lead to biased estimates, such as

attenuation bias in simple linear regression (Frost & Thompson, 2000). Recent research integrates EIV concepts with modern machine learning phenomena observed in high-dimensional settings. For example, it is demonstrated in (Kausik et al., 2023) that test error curves can exhibit double descent patterns even under distribution shift, with feature noise acting as an implicit regularizer.

## 3. Problem Setting

**Notations**   We use lower-case and upper-case letters (e.g., $x$ and $X$) to represent vectors and matrices respectively. The entries of the vectors and matrices are denoted by $x_i$ and $X_{ij}$, respectively. For a given matrix $X \in \mathbb{R}^{n \times d}$, we denote its singular values in a decreasing order $\lambda_{1,n} \geq \lambda_{2,n} \geq \cdots \geq \lambda_{\min(d,n),n} \geq 0$. Their limits, as $n, d \to \infty$, are denoted as $\lim_{n \to \infty} \lambda_{i,n} = \lambda_i$. Assume that $\lambda_{\min}(X)$ denotes the minimum singular value of matrix $X$. The aspect ratio $c_n$ for the above matrix is defined by $c_n := d/n$ and the corresponding limit is $\lim_{n \to \infty} c_n = c$. We denote the univariate and multivariate Gaussian distribution by $\mathcal{N}(\mu, \sigma^2)$ and $\mathcal{N}(\mu, \Sigma)$ respectively. We use $I_n$ to denote the identity matrix of $n$ dimension. Assume $()^+$ to be the Moore-Penrose inverse of the matrix. Let $\odot$ denote the element-wise product.

**Problem formulation**   To theoretically investigate the advantages of MoEs, particularly in the context of activation sparsity and resilience to noise, we consider a simplified linear model. This model aims to capture the essence of how MoEs might effectively utilize underlying sparse structures within data or activations, even when these structures are obscured by feature noise. This setting provides a tractable framework for comparing dense versus sparse (MoE-like) estimation strategies.

## 3.1. Why linear models?

Our adoption of a simplified linear model is a deliberate methodological choice, enabling a tractable yet insightful analysis for several key reasons.

**Theoretical tractability.** From a theoretical standpoint, starting with a linear model is a common and established practice in deep learning theory and has led to impactful insights, including the discovery of the double descent phenomenon (Belkin et al., 2019), implicit biases (Jacobs et al., 2025), Neural Tangent Kernels (Jacot et al., 2018), or mean field theory (Sirignano & Spiliopoulos, 2019). It allows us to isolate the core mechanism—how activation sparsity confers robustness to feature noise—in a theoretically manageable manner.

**Relevance to finetuning.** Furthermore, our linear setup finds strong parallels in the finetuning of LLMs through the lens of the *linear representation hypothesis*, which posits that features within LLMs are often linearly represented (Mikolov et al., 2013). This has given rise to *linear probing*, a standard technique where simple linear models are trained on the internal activations of a frozen LLM to assess the information they contain (Gurnee et al., 2023; Jiao et al., 2024). In this context, our MoE framework can be viewed as a system of specialized linear probes, where the gating mechanism routes an input to the most appropriate probe (expert) based on its internal features. This perspective grounds our work in the practical and widely-used methods for interpreting and fine-tuning large-scale models. To validate this statement, we run experiments on MoE-based probing for classification tasks. Our results, discussed in Section 5 and detailed in Appendix B.3, demonstrate that MoE probes not only achieve competitive performance but also exhibit superior robustness against feature noise compared to dense baselines, aligning with our theoretical findings.

**Empirical validation in non-linear settings.** To ensure our findings generalize beyond the linear case, we conducted controlled experiments on a synthetic dataset using a non-linear, two-layer MoE network with ReLU activations. We evaluated both regression and classification tasks under noisy conditions. The results, detailed in Appendix B.5, consistently show that the non-linear MoE models are more robust to noise than their dense counterparts (see Table 8 and Table 9). This suggests that the robustness gains we identify stem fundamentally from the modular structure of the features, an insight that holds even in more complex, non-linear architectures.

## 3.2. Linear models with block structure

Let $\beta^\star = [\beta_1^{\star T}, \ldots, \beta_k^{\star T}]^T \in \mathbb{R}^d$ be the ground truth parameter vector, partitioned into $k$ blocks corresponding to $k$ distinct "experts". The output vector $Y = [y_1, \ldots, y_n]^T \in \mathbb{R}^n$, comprising $n$ samples, is generated by:

$$Y = X\beta^\star, \quad X = \begin{bmatrix} X_1 & 0 & \ldots & 0 \\ 0 & X_2 & \ldots & 0 \\ \vdots & \vdots & \ddots & \vdots \\ 0 & 0 & \ldots & X_k \end{bmatrix}, \quad (1)$$

where $X \in \mathbb{R}^{n \times d}$ is the noiseless design matrix. We assume that $X$ possesses a block-diagonal structure, where each $X_i \in \mathbb{R}^{n_i \times d_i}$ (with $\sum n_i = N'$ for some $N'$ samples actually contributing to specific experts, and $n$ rows in $X$ formed by appropriate zero-padding for samples not pertinent to an expert block, or $\sum d_i = d$ if $X_i$ are sub-matrices of features for all $n$ samples). This block-diagonal structure represents an idealized scenario where distinct input features (or latent representations) are processed by distinct experts. This is analogous to the goal of MoEfication, where one seeks to identify or create such specialized, sparsely activated pathways from a dense network. $X_i$ can be fixed or random; for instance, its elements might be sampled from a Gaussian distribution, reflecting empirical observations in network activations (Liu et al., 2025).

In practice, we only observe a noisy version of the design matrix, $\bar{X} = X + E$, where $E = [E_{ij}]$ with $E_{ij} \sim \mathcal{N}(0, \sigma^2)$ representing feature noise. This noise can be interpreted not only as direct perturbations to input features but also as a proxy for the interference or lack of clear modularity in observed activations within a dense network before techniques like pruning reveal underlying sparse structures. We still have access to the true output $Y$.

We compare two approaches to estimate $\beta^\star$:

1. **Dense Estimator**: This approach uses the full noisy design matrix $\bar{X}$, analogous to a single, dense model attempting to learn all expert specializations simultaneously. The minimum norm estimator is:

$$\hat{\beta} = \underset{\beta}{argmin} \left\{ ||\beta||_2^2 \Big| \beta \in \underset{\beta}{argmin} \, ||Y - \bar{X}\beta||_2^2 \right\}$$
$$= \left( \bar{X}^T \bar{X} \right)^+ \bar{X}^T Y$$

2. **Sparse Estimators (MoE-like)**: Assuming a perfect gating mechanism (justified if signal-to-noise is large), we can isolate the estimation for each expert. For the $i$-th expert, using its corresponding data block $Y_i$ (rows of $Y$ corresponding to $X_i$) and noisy features $\bar{X}_i$ (the $i$-th block of $\bar{X}$ corresponding to $X_i$), the estimator is:

$$\hat{\beta}_i = \left( \bar{X}_i^T \bar{X}_i \right)^+ \bar{X}_i^T Y_i = \left( \bar{X}_i^T \bar{X}_i \right)^+ \bar{X}_i^T (X_i \beta_i^\star)$$

This decomposition into $\hat{\beta}_i$ models the behavior of an MoE where each expert focuses on a specific sub-problem. The challenge of recovering $Y_i$ or $X_i \beta_i^\star$

from noisy inputs $\bar{X}_i$ mirrors the practical scenario where an MoE, derived from a dense model, aims to reconstruct the dense model's (or an idealized target's) behavior using its specialized experts. This perspective connects to knowledge distillation, where the MoE's output (based on $\hat{\beta}_i$) is trained to match the output of a teacher (related to $Y$ or $X\beta^\star$).

Direct analysis of these minimum norm estimators under feature noise is challenging. Therefore, we first analyze their Bayes optimal counterparts, assuming access to the data distribution, to understand their performance limits with infinite data points. Subsequently, we show that MoE estimators exhibit a faster convergence speed (via gradient descent) compared to dense models in Theorem 4.7. Our hypothesis on sample complexity also suggests that MoEs have a faster convergence speed in terms of excess risk. These imply that MoEs can converge to the optimal estimator more quickly, making the analysis of the optimal estimator practically relevant.

The Bayes optimal regressor $f^\star$ minimizes the mean squared error:

$$f^\star := \underset{f}{argmin}\ \mathbb{E}_{(x,y)\sim\mathcal{D}}\left[(f(x)-y)^2\right].$$

Given our multi-expert setup, where data $(x,y)$ might arise from different underlying distributions $\mathcal{D}_i$ for each expert $i \in [k]$, we introduce a latent variable $z$ indicating the active expert:

$$f^\star := \underset{f}{argmin}\ \sum_{i=1}^{k}\mathbb{P}(z=i)\mathbb{E}_{(x,y)\sim\mathcal{D}_i}\left[(f(x)-y)^2|z=i\right].$$

(2)

For linear regressors $f(x) = x^T\beta$, the Bayes optimal estimators for the dense case, $\beta_{Dense}^{Bayes}$, and for the $i$-th sparse expert, $\beta_{Sparse,i}^{Bayes}$, are derived from minimizing this objective. The derived Bayes optimal estimators are:

$$\beta_{Dense}^{Bayes} = \begin{bmatrix} p_1(p_1\Sigma_1 + \sigma^2 I)^{-1}\Sigma_1\beta_1^\star \\ \vdots \\ p_k(p_k\Sigma_k + \sigma^2 I)^{-1}\Sigma_k\beta_k^\star \end{bmatrix},$$

(3)

$$\beta_{Sparse,i}^{Bayes} = (\Sigma_i + \sigma^2 I)^{-1}\Sigma_i\beta_i^\star, \quad i = 1,\ldots,k.$$

where $\Sigma_i$ is the covariance matrix of the noiseless features $x$ pertinent to expert $i$, $p_i = \mathbb{P}(z=i)$ is the probability of selecting expert $i$, and $\sigma^2$ relates to the variance of the feature noise $E$. These forms arise from a Bayesian linear regression perspective under Gaussian assumptions for features and noise.

## 4. MoEs Can Handle Feature Noise

This section presents our main theoretical findings, establishing the advantages of MoE-like sparse estimators over dense estimators in the context of feature noise. We analyze generalization error, robustness to perturbations, training convergence speed and sample complexity for the excess risks. All proofs are provided in the appendix.

### 4.1. Routing Simplifies to Clustering

Traditional MoE models face the challenge of learning a router, which reduces to clustering in the context of MoEfi-cation or a clear block structure. For that reason, a Bayes optimal estimate of the router can be regarded as nearly perfect, as we establish in the following.

**Decoupled training as a supervised task.** Unlike traditional MoEs where the router and experts are trained jointly from scratch in a complex optimization landscape, our approach first identifies expert structures by clustering neurons in a pre-trained dense model. This initial step provides pre-defined, ground-truth "labels" for each data point, indicating which expert it belongs to. Consequently, training the router is no longer a difficult joint optimization problem. Instead, it becomes a standard, well-posed supervised classification task: learning to predict the correct expert for a given input. This task is significantly more tractable.

**Generalization to joint optimization case.** Furthermore, our focus on the intrinsic structure of experts offers critical insights into the general joint optimization setting. Recent theoretical advancements regarding the feature learning dynamics of MoEs suggest that the router's optimization is not independent but is rather "guided" by the experts (Liao & Kyrillidis, 2026). Specifically, analyses of gradient flow dynamics demonstrate that expert parameters tend to recover the underlying data features prior to the convergence of the gating network. The router subsequently aligns its weights based on the signal provided by these specialized experts, with the experts' recovery effectively leading the router's recovery. Therefore, by explicitly characterizing expert structures in our initial phase, we capture the primary driving force of the MoE's functional partitioning. This suggests that our findings regarding expert structural advantages remain valid and explanatory even within the context of fully joint optimization.

**Theoretical guarantees.** The geometric separation of data in our modular model makes this classification task straightforward. We provide a theoretical analysis showing that a simple and efficient classifier can achieve near-perfect routing accuracy with a practical number of samples.

**Theorem 4.1** (Informal). *Under the assumption of a modular data structure (1), a Quadratic Discriminant Analysis (QDA) based router achieves an excess risk of less than $\epsilon$*

*with high probability for $n \geq \mathcal{O}(poly(d, \log(1/\delta)))$ samples.*

This result demonstrates that achieving a nearly perfect router is theoretically feasible within our framework. The formal statement of Theorem 4.1 and its complete proof are provided in Appendix A.7. This decouples the analysis of expert performance from the challenge of routing, allowing us to gain clearer insights into the inherent strengths of the expert structure itself.

## 4.2. Generalization

The following theorem compares the generalization errors of the Bayes optimal estimators for the dense and sparse cases, as defined in Eq. (3).

**Theorem 4.2.** *Consider the linear model defined in Eq. (1) and the Bayes optimal estimators $\beta_{Dense}^{Bayes}$ and $\beta_{Sparse,i}^{Bayes}$ Eq. (3). The corresponding generalization errors are:*

$$
\begin{aligned}
\mathcal{R}(\beta_{Sparse}^{Bayes}) &= \sum_{i=1}^{k} p_i \sigma^2 \beta_i^{\star T} \Sigma_i (\Sigma_i + \sigma^2 I)^{-1} \beta_i^{\star}, \\
\mathcal{R}(\beta_{Dense}^{Bayes}) &= \sum_{i=1}^{k} p_i \sigma^2 \beta_i^{\star T} \Sigma_i (p_i \Sigma_i + \sigma^2 I)^{-1} \beta_i^{\star}.
\end{aligned}
\tag{4}
$$

A key implication of Theorem 4.2 is that sparse estimators achieve better generalization performance than the dense estimator. Since $0 < p_i \leq 1$ and $\Sigma_i$ is positive semi-definite, $p_i \Sigma_i \preceq \Sigma_i$. Consequently, $p_i \Sigma_i + \sigma^2 I \preceq \Sigma_i + \sigma^2 I$. For positive definite matrices $A \preceq B$, it holds that $B^{-1} \preceq A^{-1}$. Thus, $(\Sigma_i + \sigma^2 I)^{-1} \preceq (p_i \Sigma_i + \sigma^2 I)^{-1}$. This means each term in the sum for $\mathcal{R}(\beta_{Sparse}^{Bayes})$ is less than or equal to the corresponding term for $\mathcal{R}(\beta_{Dense}^{Bayes})$, leading to $\mathcal{R}(\beta_{Sparse}^{Bayes}) \leq \mathcal{R}(\beta_{Dense}^{Bayes})$.

## 4.3. Robustness to Input Noise

Theorem 4.2 also provides a basis for understanding out-of-distribution generalization, or robustness to input perturbations. We consider two scenarios for perturbations:

1) **Perturbations not affecting routing:** The router assigns perturbed inputs to the correct experts.

2) **Perturbations causing mis-routing:** The router assigns perturbed inputs to incorrect experts.

The following theorem addresses the first scenario.

**Theorem 4.3.** *Assume input perturbations are modeled as Gaussian noise with variance $\sigma_o^2$, and the router consistently makes correct expert assignments. The generalization errors under such perturbations for the dense and sparse estimators are:*

$$
\begin{aligned}
\mathcal{R}_{\text{Dense}}(\sigma_o^2) &= \sum_{i=1}^{k} p_i \sigma^2 \beta_i^{\star\top} \Sigma_i (p_i \Sigma_i + \sigma^2 I)^{-1} \beta_i^{\star} \\
&+ \sum_{i=1}^{k} p_i^2 (\sigma_o^2 - \sigma^2) \beta_i^{\star\top} \Sigma_i (p_i \Sigma_i + \sigma^2 I)^{-2} \\
&\quad \times \Sigma_i \beta_i^{\star},
\end{aligned}
$$

$$
\begin{aligned}
\mathcal{R}_{\text{Sparse}}(\sigma_o^2) &= \sum_{i=1}^{k} p_i \sigma^2 \beta_i^{\star\top} \Sigma_i (\Sigma_i + \sigma^2 I)^{-1} \beta_i^{\star} \\
&+ \sum_{i=1}^{k} p_i (\sigma_o^2 - \sigma^2) \beta_i^{\star\top} \Sigma_i (\Sigma_i + \sigma^2 I)^{-2} \\
&\quad \times \Sigma_i \beta_i^{\star}.
\end{aligned}
$$

Sparse estimators can exhibit improved robustness (i.e., $\mathcal{R}_{Sparse}(\sigma_o^2) \leq \mathcal{R}_{Dense}(\sigma_o^2)$) under these conditions, particularly when $\sigma_o^2 > \sigma^2$. A sufficient condition for this, building upon the baseline advantage from Theorem 4.2, is if the impact of the additional error term $(\sigma_o^2 - \sigma^2)$ is less detrimental for the sparse estimators. For instance, if $\lambda_{\min}(\Sigma_i) > 4\sigma^2$ (indicating a sufficiently high signal-to-noise ratio for each expert's features) and $\sigma_o^2 > \sigma^2$, sparse estimators are favored.

However, if perturbations are large enough to cause mis-routing, the situation can change. The following theorem considers a specific mis-routing scenario.

**Theorem 4.4.** *Consider an input originally intended for expert $i$, $x_i \sim \mathcal{N}(0, \Sigma_i)$, but a perturbation of the form $\eta x_j$ (where $x_j \sim \mathcal{N}(0, \Sigma_j)$, $\eta > 1$) is introduced, causing the router to select expert $j$ with high probability. Let the overall noisy input be composite, denoted abstractly as $[0, \ldots, 0, x_i^T, 0, \ldots, 0, \eta x_j^T, 0, \ldots, 0]^T + e$, where $e \sim \mathcal{N}(0, \sigma^2 I_d)$. The generalization errors for the dense estimator:*

$$
\begin{aligned}
\mathcal{R}_{Dense}^{mis\text{-}route} &= \eta^2 p_j \beta_j^{\star T} \Sigma_j (p_j \Sigma_j + \sigma^2 I_d)^{-1} \Sigma_j \beta_j^{\star} \\
&+ \sigma^2 \eta^2 (p_j^2 - p_j) \beta_j^{\star T} \Sigma_j (p_j \Sigma_j + \sigma^2 I_d)^{-2} \Sigma_j \beta_j^{\star} \\
&- p_i \beta_i^{\star T} \Sigma_i (p_i \Sigma_i + \sigma^2 I_d)^{-1} \Sigma_i \beta_i^{\star} \\
&+ \sigma^2 \sum_{r \neq i,j}^{k} p_r^2 \beta_r^{\star T} \Sigma_r (p_r \Sigma_r + \sigma^2 I_d)^{-2} \Sigma_j \beta_r^{\star}
\end{aligned}
$$

*And for the sparse (MoE-like) estimator:* $\mathcal{R}_{Sparse}^{mis\text{-}route} = \eta^2 \beta_j^{\star T} \Sigma_j (\Sigma_j + \sigma^2 I_d)^{-1} \Sigma_j \beta_j^{\star}$.

*Remark* 4.5. The expressions in Theorem 4.4 are complex to compare directly. However, they illustrate that the performance dynamics can shift under mis-routing. In certain situations, such as when $p_r = 0$ for $r \neq i, j$ (i.e., only experts $i$ and $j$ have non-zero selection probabilities), the

dense estimator might handle such specific perturbations more effectively than a sparse estimator that is forced to use a highly specialized but incorrect expert. This suggests a trade-off: specialized experts excel when routing is correct but can be detrimental if routing fails significantly.

### 4.4. Convergence Speed

Next, we analyze the convergence dynamics of gradient descent algorithms for learning the dense and sparse estimators. We make the following simplifying assumptions:

**Assumption 4.6.** For the design matrix $X$:

i) Each $X_i$ is fixed and of size $\frac{n}{k} \times \frac{d}{k}$.

ii) The asymptotic aspect ratio $c = \lim_{n,d \to \infty} d/n > 1$. The singular values $\lambda_{ij,n}$ of $X_i$ (for $j = 1, \ldots, n/k$) converge to $\lambda_{ij}$ as $n, d \to \infty$.

iii) For all $i, j$, $\lambda_{ij} > \sqrt{c}\sigma^2$. This condition relates to signal strength versus noise level.

Under these assumptions, the following theorem characterizes the asymptotic convergence rates.

**Theorem 4.7.** *Under Assumption 4.6, the convergence rate (error reduction factor per iteration) for the $i$-th sparse estimator and dense estimator using gradient descent is given by:*

$$\rho_{Sparse,i} = 1 - \frac{\lambda_{i1}^2 \left(\sigma^2 + \lambda_{i\frac{n}{k}}^2\right)\left(c\sigma^2 + \lambda_{i\frac{n}{k}}^2\right)}{\lambda_{i\frac{n}{k}}^2 (\sigma^2 + \lambda_{i1}^2)(c\sigma^2 + \lambda_{i1}^2)},$$

$$\rho_{Dense} = 1 - \frac{\max_j \lambda_{j1}^2\left(\sigma^2 + \min_l \lambda_{l\frac{n}{k}}^2\right)\left(c\sigma^2 + \min_l \lambda_{l\frac{n}{k}}^2\right)}{\min_l \lambda_{l\frac{n}{k}}^2\left(\sigma^2 + \max_j \lambda_{j1}^2\right)\left(c\sigma^2 + \max_j \lambda_{j1}^2\right)}$$

(5)

*Typically, $\rho_{Sparse,i} \le \rho_{Dense}$, which implies faster convergence for sparse estimators. At most one sparse estimator (the one whose singular value spectrum matches the terms defining $\rho_{Dense}$) will have a convergence rate equal to that of the dense estimator; others will generally converge faster.*

The theorem suggests that decomposing the learning problem into sparse sub-problems can theoretically accelerate gradient descent. Our experimental results, involving the training of MiniMind (Gong, 2024) language models from scratch (Figure 2), corroborate this. The MoE model demonstrates competitive, and at times superior, convergence dynamics compared to the dense baseline, effectively optimizing the training loss despite a reduced computational footprint per token. Detailed experimental settings are provided in Appendix B.4.

### 4.5. Sample Efficiency

Finally, we posit a hypothesis regarding the sample complexity of the excess risk for dense and sparse estimators:

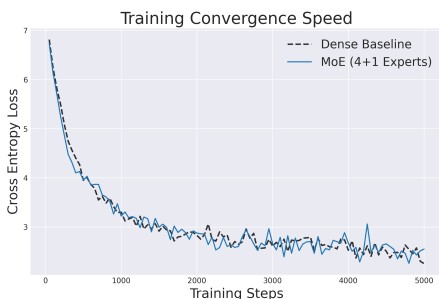

*Figure 2.* We trained both a Dense Baseline and a MoE model from scratch using the MiniMind architecture. The MoE variant employs a "Shared Expert + Routed Experts" architecture. It consists of 4 routed experts and 1 shared expert. Each token selects $K = 2$ routed experts. The intermediate dimension of each expert FFN is set to 1024. To ensure a fair comparison based on total parameter count, the intermediate dimension of the dense FFN is set to $5 \times 1024 = 5120$. This guarantees that the total parameters in the FFN layers of the dense model exactly match the sum of parameters of all experts in the MoE model. Despite having fewer active parameters per forward pass ($\sim 60\%$ of the dense model), the MoE model exhibits a training loss trajectory that closely tracks and often surpasses the convergence speed of the dense baseline, particularly in the initial phases. This aligns with Theorem 4.7, suggesting that sparse modularity facilitates efficient optimization.

*Sparse estimators achieve lower excess risk for a given sample size compared to dense estimators, implying superior sample efficiency.*

We first validate this hypothesis on our synthetic dataset constructed with a perfect modular structure. As detailed in Figure 4 and Table 4, the sparse estimator consistently exhibits significantly lower excess risk than the dense estimator throughout the training process. Interestingly, our empirical curve fitting reveals that both estimators achieve an excess risk that decays with an order of approximately $O(n^{-2})$. But the sparse estimator benefits from a much more favorable constant factor, allowing it to reach a low risk considerably faster than the dense counterpart.

This observation elucidates why a complete theoretical derivation remains challenging. Standard theoretical frameworks often rely on distinguishing estimators by their convergence rates. Since both our dense and sparse estimators converge at the same order of $O(n^{-2})$ in this noisy feature regime, such comparison methods become inapplicable. But we have some arguments from a bias-variance perspective to provide some intuitions. For a given input, the ground truth parameter vector effectively lies in a lower-dimensional subspace (e.g., $s$-dimensional) for a sparse expert, compared to the full $d$-dimensional space for the dense estimator ($s < d$). The dense estimator's attempt to learn signals in the extraneous $d - s$ dimensions can introduce larger bias. Regarding variance, sparse estimators are affected by noise primarily within their relevant $s$-dimensional subspace, while the

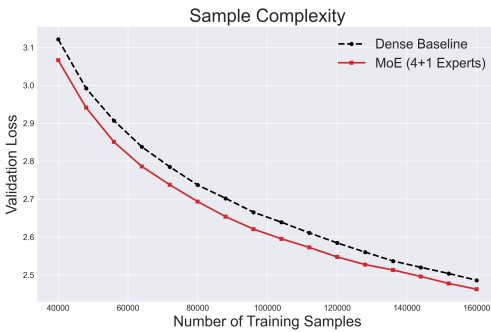

*Figure 3.* The MoE model achieves a lower validation loss faster (i.e., with fewer training samples) than the Dense Baseline. The gap highlights the superior sample efficiency of MoE.

dense estimator's variance is influenced by noise in the full $d$-dimensional space, leading to higher variance.

We further support our hypothesis by training Minimind (Gong, 2024) language models from scratch, following the configuration in Section 4.4. Our experiments demonstrate that sparse modularity effectively translates to pretraining efficiency, allowing MoEs to achieve better sample complexity even with controlled total capacity, as shown in Figure 3. See Appendix B.4 for implementation details.

## 5. Experiments

**Modular structures**   First, we provide empirical support for a key assumption in our theoretical analysis: a block-diagonal feature matrix structure module feature noise. Figure 1 demonstrates that the latent feature space of large language models (LLMs) can exhibit a similar modular, block-diagonal structure. Additional visualizations are provided in the appendix. Pretrained LLMs learn highly transferable features, requiring only a few epochs of finetuning to perform well on downstream tasks. Since these features are transferable, they can be investigated to better understand relatively general properties of activations. Specifically, we analyze the activations of intermediate layers of the Llama-2 7B model (Touvron et al., 2023) for tokens of the WikiText2 dataset (Merity et al., 2017). To highlight the block structure, we employ the recently proposed activation sparsity method TEAL (Liu et al., 2025) that removes low-magnitude activations. While these pruned activations are not necessarily noise, setting them to zero has only little impact on model performance.

**Robustness to noise**   To validate our theory that MoE can handle feature noise better, we evaluate MoE-based linear probes—interpreting MoE as a system of specialized probes (Section B.3)—against multiple strong dense baselines (Lasso, Ridge, Elastic Net) on T5-small activations.

Including Ridge and Elastic Net as baselines is important: it allows us to distinguish whether the robustness benefit of MoEs stems from modular sparsity specifically, or more broadly from regularization. As shown in Table 1, MoE configurations exhibit significantly smaller performance drops under Gaussian feature noise for most settings, particularly at high noise intensities ($\sigma \geq 1.0$). These results confirm that activation sparsity—rather than regularization alone—effectively acts as a structural noise filter on internal representations. This robustness gain generalizes to realistic input perturbations (word swap or character error) and full-model MoEfication; detailed experimental setups and expanded results are provided in Appendix B.3 and B.2.

*Table 1.* Performance drop ↓ (%) (lower is better) under high-intensity Gaussian noise ($\sigma = 2.0$) on T5-small activations. MoE consistently outperforms nearly all regularized dense baselines, demonstrating that modular sparsity—rather than regularization alone—provides the robustness benefit.

| Dataset | Lasso | Ridge | Elastic Net | MoE |
|---------|-------|-------|-------------|-----|
| SST-2   | 10.78 | 12.27 | 10.55       | **8.60** |
| CoLA    | 10.25 | 12.39 | 12.19       | **7.67** |
| MNLI    | 11.61 | 10.45 | 9.31        | **7.98** |
| AG News | 5.19  | 5.98  | **3.92**    | 7.75 |

**Robustness to approximate block structure**   A key assumption underlying our theory is that the feature matrix admits a block-diagonal structure. In practice, this structure is only approximate, as expert features may partially overlap. To assess the robustness of our mechanism to such violations, we conduct a controlled sensitivity analysis. We introduce cross-block overlap by adding a nuisance signal of magnitude $\alpha$ to all feature dimensions before applying observation noise, gradually blending the idealized modular structure into a denser one. We compare a global dense linear classifier against a routed sparse estimator (with oracle routing to isolate structural effects from optimization) as $\alpha$ varies.

*Table 2.* Classification accuracy of dense and sparse estimators under increasing cross-block feature overlap $\alpha$. Despite the departure from perfect modularity, the sparse (MoE-like) estimator maintains a consistent advantage across the full range.

| Overlap $\alpha$ | Dense Acc. | Sparse Acc. | Gap (Sparse − Dense) |
|------------------|------------|-------------|----------------------|
| 0.00 | 0.6059 | 0.7025 | +0.0966 |
| 0.05 | 0.6063 | 0.7038 | +0.0974 |
| 0.10 | 0.6054 | 0.7036 | +0.0982 |
| 0.20 | 0.6023 | 0.7021 | +0.0998 |
| 0.30 | 0.5987 | 0.6991 | +0.1004 |
| 0.50 | 0.5921 | 0.6894 | +0.0973 |
| 1.00 | 0.5659 | 0.6571 | +0.0912 |

As shown in Table 2, the sparse estimator consistently out-

performs the dense baseline across the entire range of $\alpha$. Critically, the accuracy gap remains stable and does not collapse even at high overlap ($\alpha = 1.0$), indicating that the noise-filtering advantage degrades gradually rather than disappearing abruptly when perfect block structure is violated. This confirms that the theoretical mechanism identified in Theorem 4.2 is robust to moderate generative mismatch—a realistic scenario for modern LLMs—and that realistic settings should be understood as lying between the idealized dense and sparse extremes.

**End-to-end validation on a large-scale benchmark** To confirm that the noise-filtering advantage of MoEs extends beyond synthetic data and linear probing to end-to-end trained architectures, we evaluate a **Dense ViT-L/16** (Dosovitskiy et al., 2021) against a **Sparse V-MoE-B/16** (Riquelme et al., 2021) on the ImageNet-C benchmark (Hendrycks & Dietterich, 2019) under Gaussian corruption. To ensure a fair comparison, both models are trained with state-of-the-art augmentation recipes (AugReg for ViT-L and the strong augmentation recipe for V-MoE).

*Table 3.* Top-1 accuracy (%) on ImageNet-C under Gaussian noise at severities 1, 3, and 5. Despite similar clean accuracy, the sparse V-MoE yields progressively larger robustness gains as noise severity increases. The V-MoE uses only 37% of the active parameters of ViT-L (114M vs. 307M).

| Model | Clean | Sev. 1 | Sev. 3 | Sev. 5 |
|---|---|---|---|---|
| Dense ViT-L/16 (224px) | 84.33 | 80.65 | 74.19 | 46.57 |
| Sparse V-MoE-B/16 (384px) | 84.82 | 82.35 | 77.91 | 59.65 |
| MoE Advantage ($\Delta$) | **+0.49** | **+1.70** | **+3.72** | **+13.08** |

The results in Table 3 are consistent with our theoretical predictions. While clean-data performance is nearly identical ($\Delta = +0.49\%$), the robustness gap widens dramatically as noise severity increases, reaching $+13.08\%$ at severity 5. Notably, the V-MoE achieves these gains while activating only 37% of the parameters of the dense model per forward pass, confirming that the advantage stems from sparse modular routing rather than from higher total capacity.

## 6. Conclusions

Up to our knowledge, our analysis is the first to study MoEs under feature noise, unmasking a novel mechanism underpinning the success of MoE models. We demonstrate that MoEs can outperform their dense counterparts by effectively utilizing activation sparsity in the presence of feature noise, even without an increase in total parameter size. This inherent structural advantage translates into improved generalization, enhanced robustness to input perturbations, faster training convergence, and better sample complexity. Our results provide a theoretical foundation for understanding why *modular and sparsely activated* predictors can be robust in

high-dimensional settings, and they suggest practical benefits for routed predictors such as MoE-based linear probes on frozen representations and MoEfication-like methods. Experiments on LLMs highlight the practical value of our theoretical insights, guiding the development of more computationally efficient LLMs in future. By understanding how to leverage sparsity, MoE architectures can reduce inference costs, thereby improving the accessibility of large models and reducing their environmental footprint.

## Acknowledgements

The authors gratefully acknowledge the Gauss Center for Supercomputing e.V. for funding this project by providing computing time on the GCS Supercomputer JUWELS at Jülich Supercomputing Centre (JSC). We also gratefully acknowledge funding from the European Research Council (ERC) under the Horizon Europe Framework Programme (HORIZON) for proposal number 101116395 SPARSE-ML.

## Impact Statement

This paper presents work whose goal is to advance the field of machine learning. There are many potential societal consequences of our work, none of which we feel must be specifically highlighted here.

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

# A. Proofs and further analysis

## A.1. Derivation of Bayes optimal estimators

To derive the Bayes optimal estimators for both the dense and sparse cases, we consider the scenario with infinite data samples where we know the distribution of the covariates, the output, and the feature noise. Under these conditions, we can compute the generalization error as:

$$
\begin{aligned}
\mathcal{R}(\hat{\beta}) &= \sum_{i=1}^{k} \mathbb{P}(z = i) \mathbb{E}_{(x,y)\sim\mathcal{D}_i} \left[ (f(x) - y)^2 \big| z = i \right] \\
&= \sum_{i=1}^{k} p_i \mathbb{E}_{x\sim D_i, e} \left[ ((x + e)^T \hat{\beta} - x^T \beta_i^\star)^2 \right] \\
&= \sum_{i=1}^{k} p_i \mathbb{E}_{x\sim D_i, e} \left[ (\beta_i^\star)^T x x^T \beta_i^\star + \hat{\beta}^T x x^T \hat{\beta} + \hat{\beta}^T e e^T \hat{\beta} - 2\hat{\beta}^T x x^T (\beta_i^\star)^T \right] \\
&= \sum_{i=1}^{k} p_i \left( (\beta_i^\star)^T \Sigma_i \beta_i^\star + \hat{\beta}_i^T \Sigma_i \hat{\beta}_i + \sigma^2 \hat{\beta}^T \hat{\beta} - 2\hat{\beta}_i^T \Sigma_i (\beta_i^\star)^T \right)
\end{aligned}
\tag{6}
$$

The third equality follows from the independence of the covariate $x$ and feature noise $e$, which makes the expectation of their cross terms equal to zero. The final equality uses the covariance matrices of $x$ and $e$.

The Bayes optimal estimator minimizes the generalization error, which requires:

$$
\nabla_{\hat{\beta}} \mathcal{R}(\hat{\beta}) \bigg|_{\hat{\beta} = \beta_{Dense}^{Bayes}} = 0
$$

Substituting Equation (6) and solving this optimization problem yields the form of $\beta_{Dense}^{Bayes}$. The Bayes sparse estimators can be obtained using the same approach.

## A.2. Proof of Theorem 4.2

The generalization error of the Bayes dense estimator is:

$$
\begin{aligned}
\mathcal{R}(\beta_{Dense}^{Bayes}) &= \sum_{i=1}^{k} p_i \left( (\beta_i^\star)^T \Sigma_i \beta_i^\star + (\beta_{Dense,i}^{Bayes})^T \Sigma_i \beta_{Dense,i}^{Bayes} + \sigma^2 (\beta_{Dense}^{Bayes})^T \beta_{Dense}^{Bayes} - 2(\beta_{Dense,i}^{Bayes})^T \Sigma_i \beta_i^\star \right) \\
&= \sum_{i=1}^{k} p_i \sigma^2 (\beta_i^\star)^T \Sigma_i (p_i \Sigma_i + \sigma^2 I)^{-1} \beta_i^\star
\end{aligned}
$$

This result is obtained by substituting $\beta_{Dense}^{Bayes}$ into Equation (6). The proof for the generalization error of sparse estimators follows similar derivations and is therefore omitted.

## A.3. Proof of Theorem 4.3

The process to obtain the robustness under correct routing for both the sparse estimators and dense estimator is similar, so we take the dense estimator as an example. The generalization error of the dense estimator is:

$$
\mathcal{R}_{Dense}(\sigma_o^2) = \sum_{i=1}^{k} p_i \mathbb{E}_{x \sim D_i, e' \sim \mathcal{N}(0, \sigma_o^2 I_d)} \left[ (\beta_i^\star)^T x x^T \beta_i^\star + (\beta_{Dense,i}^{Bayes})^T x x^T \beta_{Dense,i}^{Bayes} + (\beta_{Dense}^{Bayes})^T e' e'^T \beta_{Dense}^{Bayes} \right.
$$
$$
\left. - 2 (\beta_{Dense,i}^{Bayes})^T x x^T \beta_i^\star \right]
$$
$$
= \sum_{i=1}^{k} p_i \left( (\beta_i^\star)^T \Sigma_i \beta_i^\star + (\beta_{Dense,i}^{Bayes})^T \Sigma_i \beta_{Dense,i}^{Bayes} + \sigma_o^2 (\beta_{Dense}^{Bayes})^T \beta_{Dense}^{Bayes} - 2 (\beta_{Dense,i}^{Bayes})^T \Sigma_i \beta_i^\star \right)
$$
$$
= \sum_{i=1}^{k} p_i \sigma^2 (\beta_i^\star)^T \Sigma_i (p_i \Sigma_i + \sigma^2 I)^{-1} \beta_i^\star + \sum_{i=1}^{k} p_i^2 (\sigma_o^2 - \sigma^2)(\beta_i^\star)^T \Sigma_i (p_i \Sigma_i + \sigma^2 I)^{-2} \Sigma_i \beta_i^\star
$$

## A.4. Proof of Theorem 4.4

In the mis-routing case, under the specific perturbation, the result for sparse estimator is just $\eta^2$ scaled of the generalization error in Theorem 4.2. The generalization error for the input $[0, \ldots, 0, x_i^T, 0, \ldots, 0, \eta x_j^T, 0, \ldots, 0]^T + e$ is:

$$
\mathcal{R}_{Dense}^{\text{mis-route}} = (\beta_i^\star)^T \Sigma_i \beta_i^\star + (\beta_{Dense,i}^{Bayes})^T \Sigma_i \beta_{Dense,i}^{Bayes} + \eta^2 (\beta_{Dense,j}^{Bayes})^T \Sigma_j \beta_{Dense,j}^{Bayes}
$$
$$
+ \sigma^2 (\beta_{Dense}^{Bayes})^T \beta_{Dense}^{Bayes} - 2 (\beta_{Dense,i}^{Bayes})^T \Sigma_i \beta_i^\star
$$
$$
= \eta^2 p_j \beta_j^{\star T} \Sigma_j (p_j \Sigma_j + \sigma^2 I_d)^{-1} \Sigma_j \beta_j^\star + \sigma^2 \eta^2 (p_j^2 - p_j) \beta_j^{\star T} \Sigma_j (p_j \Sigma_j + \sigma^2 I_d)^{-2} \Sigma_j \beta_j^\star
$$
$$
- p_i \beta_i^{\star T} \Sigma_i (p_i \Sigma_i + \sigma^2 I_d)^{-1} \Sigma_i \beta_i^\star + \sigma^2 \sum_{r \neq i,j}^{k} p_r^2 \beta_r^{\star T} \Sigma_r (p_r \Sigma_r + \sigma^2 I_d)^{-2} \Sigma_j \beta_r^\star
$$

## A.5. Proof of Theorem 4.7

We first establish that the convergence rate of the least squares estimator is related to the condition number of the design matrix $X$. Using gradient descent, the update step follows:

$$
\hat{\beta}_{t+1} = \hat{\beta}_t - \eta_t \left( X^T X \hat{\beta}_t - X^T y \right) \tag{7}
$$

where $\eta_t$ is the step size.

Let the singular value decomposition of the design matrix $X$ be $U \Sigma V^T$, and choose the step size $\eta_t = \frac{1}{\lambda_1^2}$ for all $t$, where $\lambda_1$ is the maximal singular value of $X$. Then the residual follows the dynamics:

$$
X \hat{\beta}_{t+1} - y = X \hat{\beta}_t - y - \eta_t X \left( X^T X \hat{\beta}_t - X^T y \right)
$$
$$
= X \hat{\beta}_t - y - \eta_t X X^T \left( X \hat{\beta}_t - y \right)
$$
$$
= \left( I_n - \eta_t X X^T \right) \left( X \hat{\beta}_t - y \right)
$$
$$
= U \text{diag} \left( 1 - \frac{\sigma_1^2}{\sigma_1^2}, \ldots, 1 - \frac{\sigma_d^2}{\sigma_1^2} \right) U^T \left( X \hat{\beta}_t - y \right)
$$
$$
= U \text{diag} \left( \left( 1 - \frac{\sigma_1^2}{\sigma_1^2} \right)^{t+1}, \ldots, \left( 1 - \frac{\sigma_d^2}{\sigma_1^2} \right)^{t+1} \right) U^T \left( X \hat{\beta}_0 - y \right)
$$

Therefore, the convergence rate depends on $1 - \frac{\lambda_d^2}{\lambda_1^2}$. For the design matrices $\bar{X}_i = X_i + E_i$ (where $i = 1, \ldots, k$) and $\bar{X}$, the convergence rates depend on $1 - \frac{\bar{\lambda}_{d/k}^2}{\lambda_{i1}^2}$ and $1 - \frac{\bar{\lambda}_d^2}{\lambda_1^2}$, respectively.

Based on Theorem 6 in (Loubaton & Vallet, 2011), the perturbed singular values $\bar{\lambda}_i^2$ and the non-perturbed singular values $\lambda_i^2$ satisfy the following relationship as $n \to \infty$:

$$\bar{\lambda}_i^2 \to \begin{cases} \frac{(\sigma^2 + \lambda_i^2)(c\sigma^2 + \lambda_i^2)}{\lambda_i^2}, & \text{if } \lambda_i^2 > \sqrt{c}\sigma^2 \\ \sigma^2(1 + \sqrt{c})^2, & \text{otherwise} \end{cases}$$

Substituting this result into the convergence speed analysis yields Theorem 4.7.

### A.6. Analysis of sample complexity of minimum norm estimators

As mentioned in the main text, analyzing the sample complexity of minimum norm estimators presents significant challenges. The difficulty arises not only because the sample complexities of both sparse and dense estimators have the same order, but also because analyzing the excess risk of each estimator individually is inherently difficult.

The excess risk is the difference between the test error and the irreducible risk (the Bayes optimal error). We first examine the test error structure since we already have the Bayes optimal risk.

The bias and variance decomposition of the test error is:
$$\mathbb{E}_{x,e}[(x + e)^T \hat{\beta} - x^T \beta]^2 = \mathbb{E}_{x,e}[x^T(\hat{\beta} - \beta)]^2 + \mathbb{E}_e(e^T \hat{\beta})^2$$

where the first term represents the bias and the second term represents the variance.

#### A.6.1. BIAS ANALYSIS

Since $\bar{X}$ is full rank, we have:
$$\hat{\beta} - \beta^* = -(\bar{X}^T \bar{X})^+ \bar{X}^T E \beta^*$$

where $\bar{X} = X + E$. The bias term becomes:
$$\begin{aligned} \mathbb{E}_x[x^T(\hat{\beta} - \beta^*)]^2 &= (\hat{\beta} - \beta^*)^T \mathbb{E}_x[xx^T](\hat{\beta} - \beta^*) \\ &= (\hat{\beta} - \beta^*)^T \Sigma (\hat{\beta} - \beta^*) \\ &= (\beta^*)^T E^T \bar{X} (\bar{X}^T \bar{X})^+ \Sigma (\bar{X}^T \bar{X})^+ \bar{X}^T E \beta^* \end{aligned}$$

This equality holds for both sparse and dense estimators. However, this expression is difficult to analyze because $E$ and $\bar{X}$ are highly correlated.

#### A.6.2. VARIANCE ANALYSIS

For the $i$-th sparse estimator, we assume that the diagonal block $X_i$ is sampled from a Gaussian distribution $\mathcal{N}(0, \Sigma_i)$, and the covariance matrix has eigenvalue decomposition $\Sigma_i = U_i \Lambda_i U_i^T$, where $\Lambda_i = \text{diag}\{\lambda_{i1}, \dots, \lambda_{i,n/k}\}$.

$$\begin{aligned} \text{Var}(\hat{\beta}) &= \sigma^2 \text{tr}\left[ \left(\bar{X}_i^T \bar{X}_i\right)^+ \bar{X}_i^T X_i \beta_i^\star (\beta_i^\star)^T X_i^T \bar{X}_i \left(\bar{X}_i^T \bar{X}_i\right)^+ \right] \\ &= \sigma^2 \text{tr}\left[ \left(\bar{X}_i\right)^+ X_i X_i^T \left(\bar{X}_i\right)^{+T} \right] \\ &= \sigma^2 \text{tr}\left[ \left(\bar{X}_i \bar{X}_i^T\right)^+ X_i X_i^T \right] \\ &= \sigma^2 \text{tr}\left[ X_i^T \left(\bar{X}_i \bar{X}_i^T\right)^+ X_i \right] \end{aligned}$$

For $\bar{X}_i$, we have:

$$\begin{aligned} \bar{X}_i &= X_i + E_i \\ &= Z_i \Lambda_i^{1/2} U_i^T + \sigma W_i \\ &= \left(Z_i \Lambda_i^{1/2} + \sigma W_i U_i\right) U_i^T \end{aligned}$$

The elements of $Z_i$ and $W_i$ are sampled independently from standard Gaussian distributions. The last equality holds because standard Gaussian random vectors are invariant under orthonormal transformations. Then $Z_i \Lambda_i^{1/2} + \sigma W_i U_i$ is a Gaussian random matrix whose $(r, s)$-th entry is a Gaussian random variable with mean 0 and variance $\lambda_{ir}^2 + \sigma^2$.

Based on this observation, the variance term becomes:

$$\mathrm{Var}(\hat{\beta}_i) = \sigma^2 \mathrm{tr} \left[ \sum_{r=1}^{d/k} (\lambda_{ir}^2 + \sigma^2) z_r^T \left( \bar{X}_i \bar{X}_i^T \right)^+ z_r \right]$$

$$\leq \sigma^2 \lambda_{\min}^{-1} \left( \bar{X}_i \bar{X}_i^T \right) \sum_{r=1}^{d/k} (\lambda_{ir}^2 + \sigma^2) z_r^T z_r$$

We can bound $\lambda_{\min}^{-1} \left( \bar{X}_i \bar{X}_i^T \right)$ using the following lemma, since $\bar{X}_i \bar{X}_i^T = \sum_{r=1}^{d/k} (\lambda_{ir}^2 + \sigma^2) z_r z_r^T$:

**Lemma A.1** (Lemma 5 in (Li et al., 2023b)). *Let*

$$\hat{\mathbf{A}} = \sum_{i=1}^{n} \hat{\lambda}_i \mathbf{w}_i \mathbf{w}_i^T,$$

*where $\mathbf{w}_i \in \mathbb{R}^d$ is a random vector with each entry independently distributed as $\mathcal{N}(0, 1)$. There exists a universal constant $b_1$ such that with probability at least $1 - 2e^{-t}$, we have:*

$$\sum_{i=1}^{n} \hat{\lambda}_i - \Lambda \leq \mu_n(\hat{A}) \leq \mu_1(\hat{A}) \leq \sum_{i=1}^{n} \hat{\lambda}_i + \Lambda$$

*where*

$$\Lambda = b_1 \left( \hat{\lambda}_1 (t + n \log 9) + \sqrt{(t + n \log 9) \sum_{i=1}^{n} \hat{\lambda}_i^2} \right)$$

*Furthermore, there exists a universal constant $b_2$ such that with probability at least $1 - 2e^{-n/b_2}$:*

$$\frac{1}{b_2} \sum_{i=1}^{n} \hat{\lambda}_i - b_2 \hat{\lambda}_1 n \leq \lambda_{min}(\hat{A}) \leq \lambda_{max}(\hat{A}) \leq b_2 \sum_{i=1}^{n} \hat{\lambda}_i + b_2 \hat{\lambda}_1 n$$

Assume there exists a $t^\star$ satisfying:

$$t^\star = \min \left\{ 0 \leq j \leq n/k : \frac{\sum_{r=j+1}^{n/k} (\lambda_{ir}^2 + \sigma^2)}{\lambda_{i(j+1)}^2 + \sigma^2} > bn \right\}$$

for some constant $b$. Then we have:

$$\lambda_{\min} \left( \sum_{r=1}^{d/k} (\lambda_{ir}^2 + \sigma^2) z_r z_r^T \right) \geq \lambda_{\min} \left( \sum_{r=t^\star+1}^{d/k} (\lambda_{ir}^2 + \sigma^2) z_r z_r^T \right) \geq c_1 \sum_{r=t^\star+1}^{n/k} (\lambda_{ir}^2 + \sigma^2)$$

for some constant $c_1$, due to Lemma A.1.

The term $\sum_{r=1}^{d/k} (\lambda_{ir}^2 + \sigma^2) z_r^T z_r$ can also be upper bounded since $z_r^T z_r$ concentrates around $n/k$:

$$\sum_{r=1}^{d/k} (\lambda_{ir}^2 + \sigma^2) z_r^T z_r \leq c_2 n \sum_{r=1}^{d/k} (\lambda_{ir}^2 + \sigma^2)/k$$

for some constant $c_2$ with high probability.

Therefore, we obtain an upper bound for the variance term with high probability:

$$\text{Var}(\hat{\beta}_i) \le \sigma^2 c_3 \frac{n \sum_{r=1}^{d/k} (\lambda_{ir}^2 + \sigma^2)}{k \sum_{r=t^\star+1}^{n/k} (\lambda_{ir}^2 + \sigma^2)}$$

However, this upper bound is too loose as it can diverge as $n \to \infty$, while in practice the variance does not exhibit this behavior.

For the dense estimator, obtaining even such a crude bound is challenging. If we use the notation:

$$\bar{X}\bar{X}^T = \sum_{i=1}^d s_i s_i^T$$

where $s_r = \left[\sigma z_{r,1}^T \dots \sqrt{(\lambda_{ir} + \sigma^2)} z_{r,i}^T \dots \sigma z_{r,k}^T\right]$, then applying Lemma A.1 becomes impossible because we cannot extract a common scalar factor and construct columns consisting of independent and identically distributed standard Gaussian random variables.

### A.6.3. A CASE STUDY

In this section, we analyze a simplified case for sparse estimators where the number of features $d$ is equal to the number of experts $k$. This implies that each expert focuses on a single, unique dimension. We consider a one-dimensional scenario for this analysis. Assume the true feature $x \sim \mathcal{N}(0, \lambda^2)$, the observation error $e \sim \mathcal{N}(0, \sigma^2)$, and the response $y = x\beta$. The feature $x$ is observed with noise, so the regressor used is $x' = x + e$. We analyze the underparameterized case, ensuring at least one data sample per expert for estimation (here, $n \ge 1$ for the single parameter $\beta$).

The Ordinary Least Squares (OLS) estimator $\hat{\beta}$ for regressing $y_i$ on $x_i + e_i$ is:

$$\hat{\beta} = \frac{\sum_{i=1}^n (x_i + e_i)(\beta x_i)}{\sum_{i=1}^n (x_i + e_i)^2} = \beta \frac{\frac{1}{n}\sum_{i=1}^n (x_i^2 + x_i e_i)}{\frac{1}{n}\sum_{i=1}^n (x_i^2 + 2 x_i e_i + e_i^2)}$$

The generalization error is:

$$\mathcal{R} = \mathbb{E}_{x_i, e_i}\left[\lambda^2(\hat{\beta} - \beta)^2 + \sigma^2 \hat{\beta}^2\right]$$
$$= \lambda^2 \mathbb{E}[(\hat{\beta} - \beta)^2] + \sigma^2 \mathbb{E}[\hat{\beta}^2]$$

Based on the law of large numbers, the bias term $\lambda^2 \mathbb{E}[(\hat{\beta} - \beta)^2]$ can be approximated by $\frac{\sigma^2 \lambda^2 \beta^2}{\lambda^2 + \sigma^2}$, which is exactly the Bayes error in this case. Then by the Delta method, the variance is:

$$\sigma^2 \mathbb{E}[\hat{\beta}^2] = \beta^2 \left(\frac{\lambda^2}{\lambda^2 + \sigma^2}\right)^2 \frac{1}{n}\left[\frac{2\lambda^4 + \lambda^2\sigma^2}{(\lambda^2)^2} + \frac{2(\lambda^2 + \sigma^2)^2}{(\lambda^2 + \sigma^2)^2} - \frac{2 \cdot 2\lambda^2(\lambda^2 + \sigma^2)}{\lambda^2(\lambda^2 + \sigma^2)}\right]$$
$$= \sigma^2 \beta^2 \left(\frac{\lambda^2}{\lambda^2 + \sigma^2}\right)^2 \frac{1}{n}\left[\frac{2\lambda^2 + \sigma^2}{\lambda^2} + 2 - 4\right]$$
$$= \sigma^2 \beta^2 \left(\frac{\lambda^2}{\lambda^2 + \sigma^2}\right)^2 \frac{1}{n}\left[\frac{2\lambda^2 + \sigma^2 - 2\lambda^2}{\lambda^2}\right]$$
$$= \sigma^2 \beta^2 \left(\frac{\lambda^2}{\lambda^2 + \sigma^2}\right)^2 \frac{1}{n}\frac{\sigma^2}{\lambda^2}$$
$$= \frac{\beta^2 \lambda^2 \sigma^4}{n(\lambda^2 + \sigma^2)^2}$$

However, our experiments (see Figure 4 and Table 4) indicate different convergence behaviors. For the dense estimator (not analyzed here), the excess risk fits well to an $O(n^{-2})$ curve. For the sparse estimator (analyzed in this section), the empirical fit includes both $O(n^{-1})$ and $O(n^{-2})$ terms. These empirical findings suggest that higher-order terms or other phenomena

not captured by this analysis play a significant role, underscoring the challenges in obtaining a precise theoretical analysis of sample complexity that fully matches experimental results.

In Table 4, we list the closed-form fitting curves for the excess risks of both dense and sparse estimators, corresponding to the experimental conditions shown in Figure 4.

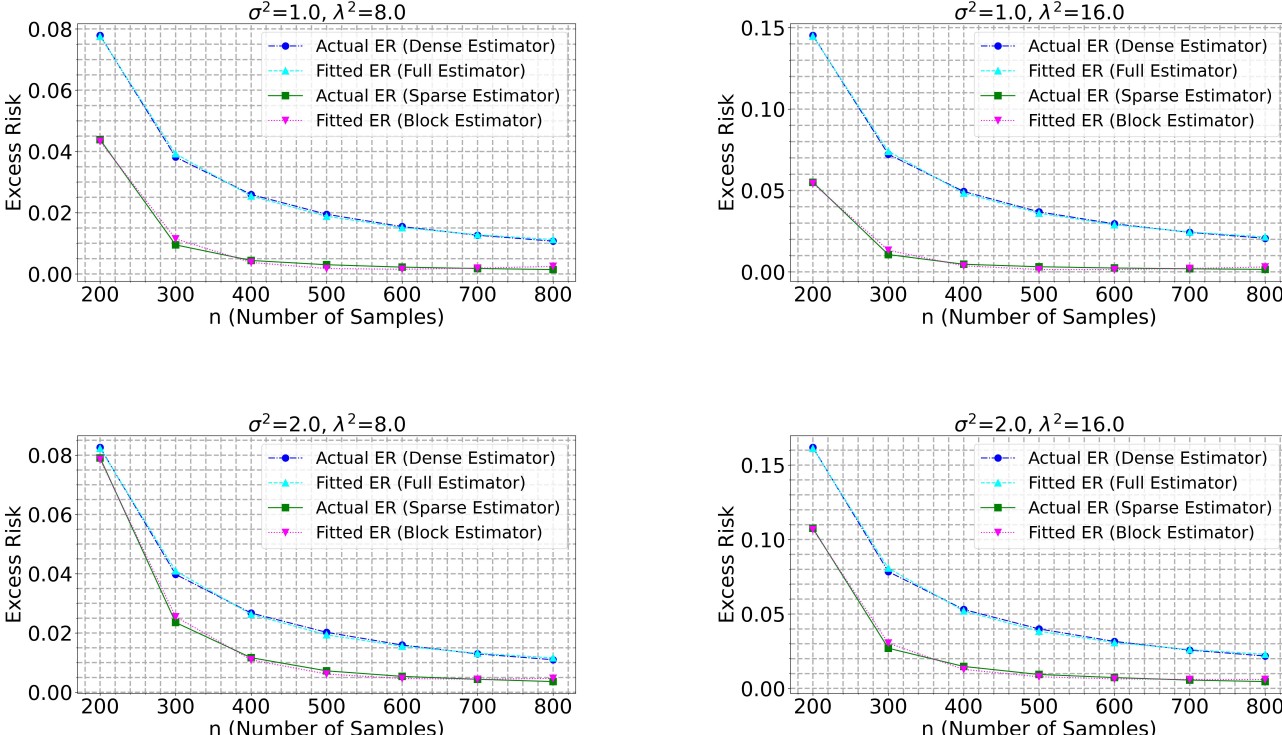

*Figure 4.* The actual and fitted curves for the excess risks of the dense estimator and sparse estimator under the setting $d = k = 100$.

*Table 4.* Closed-form fitting curves for the excess risks under the parameter setting $\sigma^2 = 1$ and $\lambda^2 = 8$. Four different runs are presented.

| Run | Dense estimator | Sparse estimator | |
|-----|-----------------|------------------|---|
| 1 | $\dfrac{2.67 \times 10^3}{n^2}$ | $\dfrac{3.94 \times 10^3}{n^2}$ | $-\dfrac{13.76}{n}$ |
| 2 | $\dfrac{4.86 \times 10^3}{n^2}$ | $\dfrac{5.24 \times 10^3}{n^2}$ | $-\dfrac{19.10}{n}$ |
| 3 | $\dfrac{2.92 \times 10^3}{n^2}$ | $\dfrac{5.51 \times 10^3}{n^2}$ | $-\dfrac{11.10}{n}$ |
| 4 | $\dfrac{5.64 \times 10^3}{n^2}$ | $\dfrac{11.7 \times 10^3}{n^2}$ | $-\dfrac{70.85}{n}$ |

## A.7. Proof of nearly perfect router

### A.7.1. FORMAL STATEMENT OF THEOREM 4.1

Consider $n$ samples arranged in matrix $\bar{X} \in \mathbb{R}^{n \times d}$ as in the paper. Assign labels $Y \in \{1, \dots, k\}^n$ indicating block membership for each row. Given training data $(\bar{X}, Y)$, we train a classifier for test points $x_{\text{test}} \in \mathbb{R}^d$, as the router. We design a quadratic discriminant analysis (QDA) based classifier for it. To train such a QDA classifier, we assume we have access to $\sigma^2$ (estimation of $\sigma^2$ is easy) and denote the feature sets for each block as $\{S_i\}_{i=1}^k$. We assume a balanced loading, i.e., $n_i \approx n_j, \forall i, j$, for simplicity.

To estimate the covariance matrix for each block $i$, we need two steps: first, extract submatrix $\bar{X}_{i,S_i} \in \mathbb{R}^{n_i \times d_i}$ (rows with $Y_j = i$, columns $S_i$); second, compute sample covariance:

$$\hat{C}_i = \frac{1}{n_i}(\bar{X}_{i,S_i})^\top \bar{X}_{i,S_i}$$

For a test point $x_{\text{test}} \in \mathbb{R}^d$, we first need to compute the discriminant score for each class $i$:

$$\hat{g}_i(x_{\text{test}}) = -\frac{1}{2}\log|\hat{C}_i| - \frac{1}{2}x_{\text{test},S_i}^\top \hat{C}_i^{-1} x_{\text{test},S_i}$$

Then the predicting label should be $\hat{y} = \arg\max_{i \in \{1,\ldots,k\}} \hat{\delta}_i(x_{\text{test}})$. Next is the sample complexity: for any $\epsilon > 0, \delta \in (0,1)$, if the total samples satisfy:

$$n \geq C\frac{M^2}{\epsilon^2}\sum_{i=1}^{k}\max\left\{d_i, \log\left(\frac{k}{\delta}\right)\right\}$$

then with probability at least $1 - \delta$, the excess risk $R - R^* \leq \epsilon$. Here $M = \max_i \|C_i\|_2 \leq \max_i \|\Sigma_i\|_2 + \sigma^2 < \infty$.

### A.7.2. PROOF

The excess risk can be upper bounded by (Fan et al., 2012):

$$R - R^* \leq K\max_i \|\hat{C}_i - C_i\|_2$$

where $K > 0$ is a constant depending on the separation degree between classes. The estimation error of the sample covariance matrix can be upper bounded by (Wainwright, 2019):

$$\mathbb{P}\left(\|\hat{C}_i - C_i\|_2 \geq c_1 M\left(\sqrt{\frac{d_i}{n_i}} + \frac{d_i}{n_i}\right) + t\right) \leq c_2 \exp\left(-c_3 n_i \min\left\{\frac{t^2}{M^2}, \frac{t}{M}\right\}\right)$$

with $n_i$ samples, where $c_1, c_2, c_3 > 0$ are absolute constants. Denote $\eta = \epsilon/K$. If $n_i \geq C_1\frac{M^2 d_i}{\eta^2}$, where $C_1$ is a large enough constant compared to $c_1$, we have $c_1 M\left(\sqrt{\frac{d_i}{n_i}} + \frac{d_i}{n_i}\right) \leq \eta/2$. If we also have $n_i \geq \frac{4M^2}{c_3\eta^2}\log(\frac{c_2 k}{\delta})$, then the following inequality holds:

$$\mathbb{P}\left(\|\hat{C}_i - C_i\|_2 \geq \eta\right) \leq \frac{\delta}{k}$$

Then by the union bound, we can get the sample complexity result. This sample complexity result means that we can get a "perfect" router with large enough dataset.

## B. Experimental details

We provide details and hyperparameters used for the experiments.

### B.1. Modular structure

The following methodology is used to generate the modular structure visualizations presented in the paper. We begin by applying the TEAL algorithm (Liu et al., 2025) to obtain sparse activations. TEAL achieves substantial activation sparsity with minimal loss to model performance by identifying layer-specific thresholds and pruning input activations with magnitudes below those thresholds. Two thresholding strategies are available to achieve a target sparsity: (1) **uniform thresholding**, where the sparsity constraint is applied uniformly across the input activations of all linear layers in the model, and (2) **greedy thresholding**, where thresholds are selected per layer to minimize performance loss while meeting an overall sparsity constraint at the decoder block level. In our experiments, we use the precomputed thresholds provided in the TEAL GitHub repository (https://github.com/FasterDecoding/TEAL). All subsequent analysis is performed on the activations pruned using one of these two approaches.

To uncover the modular structure, we start by identifying feature blocks. We compute an affinity matrix over feature dimensions using cosine similarity and apply spectral clustering to group similar features into blocks. We divide features into

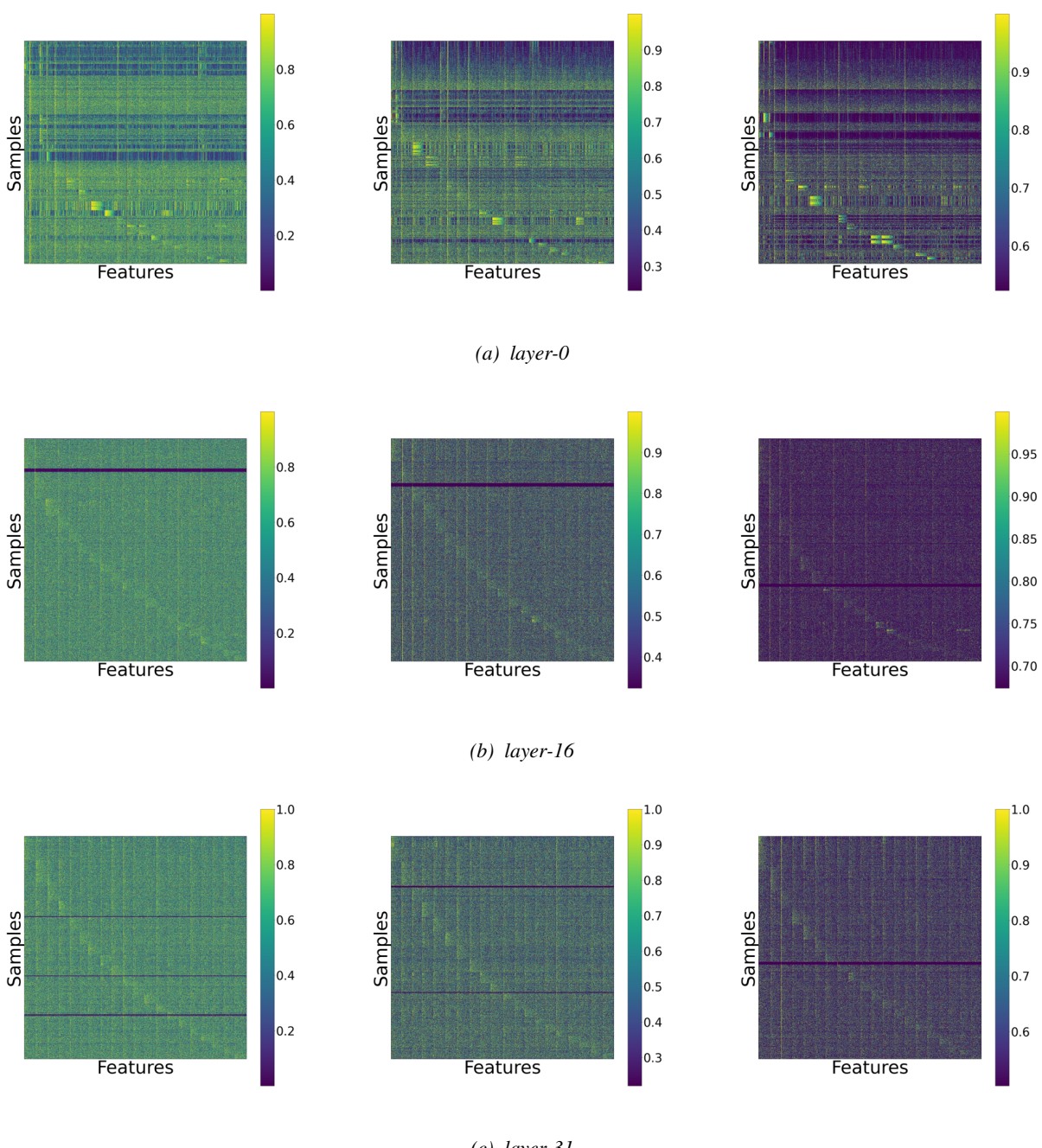

*Figure 5.* Modular structure in input activations to the `up_proj` layer within the MLP block of (a) *layer-0*, (b) *layer-16*, and (c) *layer-31* of the Llama-2-7B model, revealed using TEAL for greedy magnitude pruning. From left to right, panels show activation percentiles after pruning activations based on 0%, 40%, and 70% *greedy* activation sparsity thresholds, respectively.

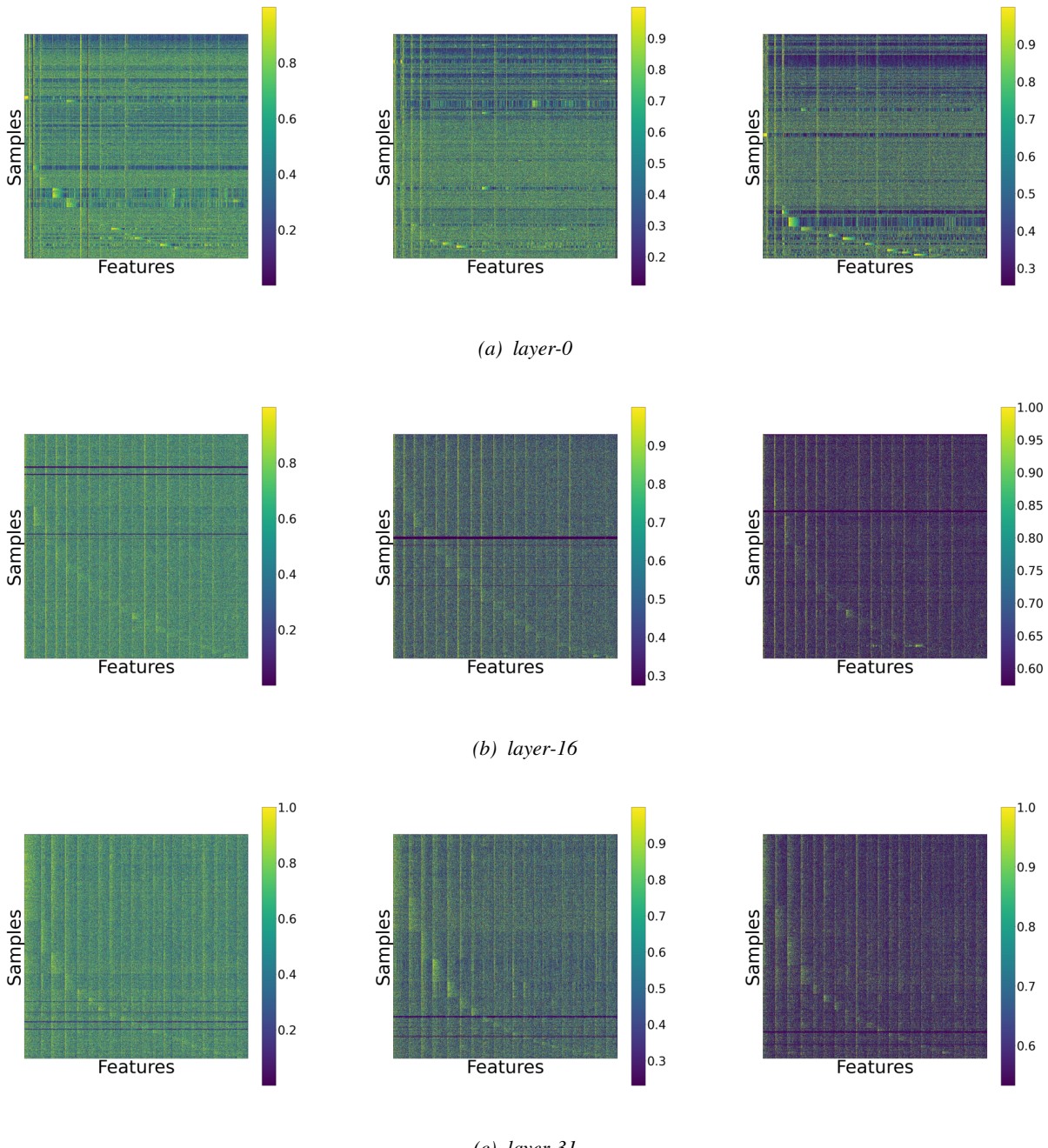

*(a) layer-0*

*(b) layer-16*

*(c) layer-31*

*Figure 6.* Modular structure in input activations to the `up_proj` layer within the MLP block of (a) *layer-0*, (b) *layer-16*, and (c) *layer-31* of the Llama-3.1-8B model, revealed using TEAL for greedy magnitude pruning. From left to right, panels show activation percentiles after pruning activations based on 0%, 40%, and 70% *greedy* activation sparsity thresholds, respectively.

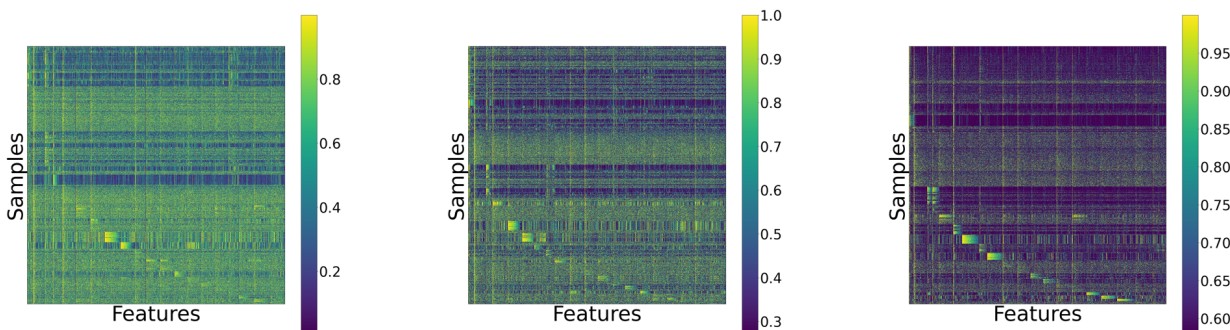

*Figure 7.* Modular structure in input activations to the `gate_proj` layer within the MLP block of *layer-0* of the Llama-2-7B model, revealed using TEAL for greedy magnitude pruning. From left to right, panels show activation percentiles after pruning activations based on 0%, 40%, and 70% *greedy* activation sparsity thresholds, respectively.

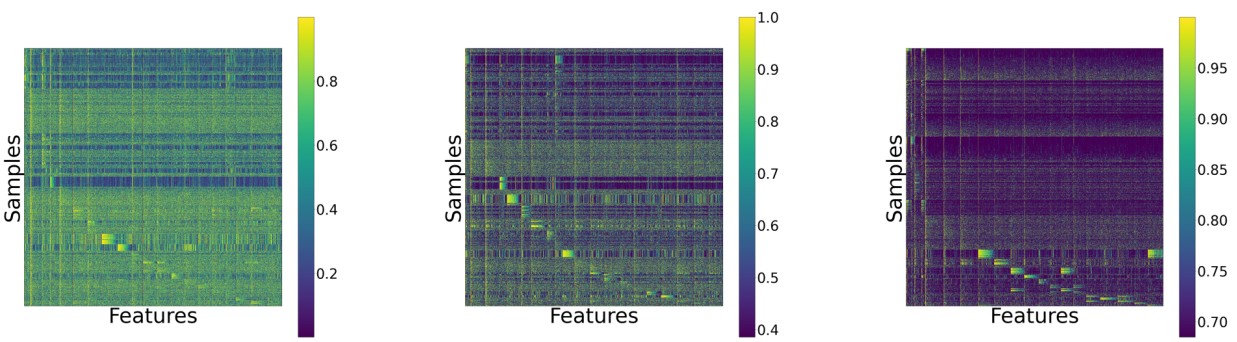

*Figure 8.* Modular structure in input activations to the `up_proj` layer within the MLP block of *layer-0* of the Llama-2-7B model, revealed using TEAL for uniform magnitude pruning. From left to right, panels show activation percentiles after pruning activations based on 0%, 40%, and 70% *uniform* activation sparsity thresholds, respectively.

20 such modular for all experiments. Tokens are then assigned to the most compatible feature modular based on activation magnitudes, and both features and tokens are reordered such that the $i^{th}$ feature modular and its associated tokens form the $i^{th}$ diagonal block of the resulting feature matrix. Token-to-modular assignment is determined by ranking activation magnitudes and assigning each token to the modular where it obtains the highest average percentile rank across the block's features. For visualization, we use these percentile ranks rather than raw activation values to reduce the visual impact of extreme outliers, which can otherwise dominate the heatmap due to their magnitude being orders above the mean.

It is important to note that feature and token orderings are recomputed independently for each configuration (i.e., layer and sparsity level). As a result, the block diagonal structures seen in different visualizations are not directly comparable: each reflects a distinct clustering and reordering based on its specific activation pattern.

Figure 1 provides initial evidence for the emergence of modular structure in the input activations to the MLP block of layer 2 in the Llama-2-7B model. In all visualizations, activations are collected by passing tokens from the validation set of the Wikitext2 (`wikitext-2-raw-v1`) dataset through the model with a fixed sequence length of 1024. This design choice is consistent across all experiments. To assess the generality of the observed modular structure, we include further visualizations in Figures 5 and 6. These show activation patterns from the `up_proj` layers of the MLP blocks at the $0^{th}$, $16^{th}$, and $31^{st}$ decoder layers of both Llama-2-7B (Touvron et al., 2023) and Llama-3.1-8B models (Grattafiori et al., 2024), using greedy thresholding at 0%, 40%, and 70% sparsity levels. Across these settings, approximate modular-diagonal patterns persist. The structure is more pronounced in earlier decoder layers and at higher sparsity levels.

We also observe that this structure holds across both thresholding strategies. Figure 8 shows that modular structure is preserved even when using uniform thresholding. Additionally, Figure 7 illustrates similar patterns in the input activations to the `gate_proj` layers, further supporting the consistency of this phenomenon.

Each experiment requires two A100 GPUs (40GB memory) to accommodate model loading and activation storage.

## B.2. Robustness to noise

To generate the sparsely activated T5-base model using MoEfication, we use k-means clustering approach over feature dimensions and divide them into 96 experts. We choose activation sparsity levels corresponding to choosing 20,40,60, and 80 experts out of the 96. The validation set of SST2 dataset (Socher et al., 2013) is used for performance evaluation similar to (Zhang et al., 2022). We use RandomWordAug (word swap) and KeyboardAug augmentations from the NLPAug library (https://github.com/makcedward/nlpaug) to simulate word and character noise. As already mentioned, we perform at least one and a maximum of two swaps per sentence to simulate word noise. To simulate character noise, one character within each of two randomly chosen words is replaced with a keyboard error. We use a single A100 (40GB) GPU to get the activation sparse model and for further robustness evaluation.

Figure 9 shows both the accuracy under noisy and clean conditions, as well as the corresponding performance gap, with confidence intervals computed over 100 random seeds (0–99). The results provide empirical support for our hypothesis that sparsely activated models are more robust to noise than dense models. At intermediate activation sparsity levels (40 experts), not only is the performance gap lower than dense models, but the sparse models also outperform the dense models in the presence of noise.

## B.3. MoE-based linear probing details and results

In the main text, we posit that our Mixture-of-Experts (MoE) framework can be interpreted as a system of specialized linear probes, routing inputs to the most appropriate linear model based on internal feature representations. To validate this perspective and demonstrate that a collection of local linear models can exhibit superior robustness compared to a global sparse linear model, we conducted experiments using T5-small encoder activations on several benchmarks: SST-2, CoLA, MNLI, and AG News.

### B.3.1. Expert construction: constrained spectral clustering

To construct experts that are not merely semantically cohesive but also task-relevant, we employ a *Constrained Spectral Clustering* approach. Standard spectral clustering relies solely on the co-activation (covariance) of neurons, which may group features that are correlated but irrelevant to the downstream classification task.

We introduce supervision into the clustering process using the **Fisher Score**, which measures the discriminative power of

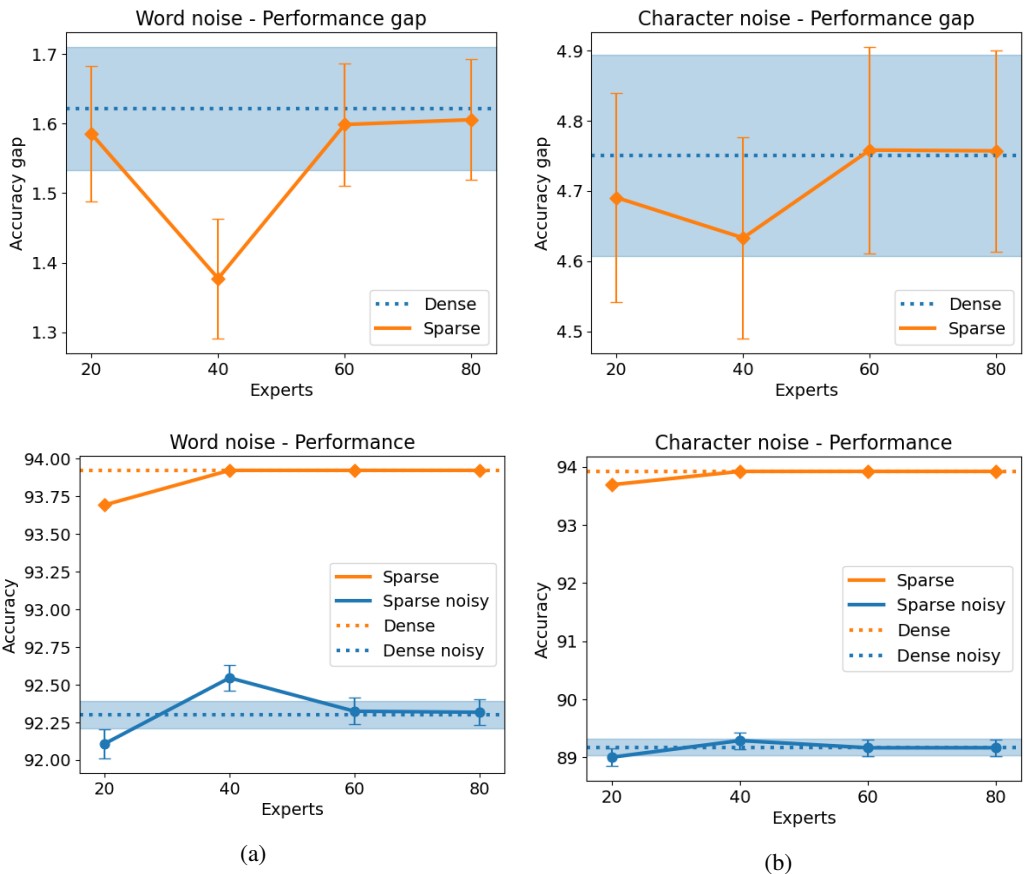

(a)                                                            (b)

*Figure 9.* Robustness to noise. The figure shows the mean performance and confidence intervals of dense and activation sparse T5-base models and the accuracy gap between them as a result of noise over SST2 dataset over 100 seeds. (a) Word noise, (b) Character noise, (top) accuracy gap of dense and sparse models over clean and noisy datasets (bottom) accuracy. The confidence intervals for sparse models over noisy dataset are shown using error bars. The same is shown for dense models using the blue color fill.

the $j$-th neuron (feature) across classes $C$. The Fisher Score for feature $j$ is calculated as:

$$FS(j) = \frac{\sum_{c=1}^{C} n_c(\mu_{j,c} - \mu_j)^2}{\sum_{c=1}^{C} n_c\sigma_{j,c}^2} \tag{8}$$

where $n_c$ is the number of samples in class $c$, $\mu_{j,c}$ and $\sigma_{j,c}^2$ are the mean and variance of feature $j$ in class $c$, and $\mu_j$ is the global mean.

We modify the affinity matrix $A$ used for spectral clustering by re-weighting the cosine similarity $S_{ij}$ between neurons $i$ and $j$:

$$A_{ij} = S_{ij} \times \exp(-|FS(i) - FS(j)|) \tag{9}$$

This constraint enforces a *must-link* tendency between neurons that share similar discriminative power, ensuring that experts are composed of features that contribute synergistically to the classification task.

### B.3.2. ROUTER TRAINING

Since our experts are trained as independent linear probes (Logistic Regression with $\ell_2$ regularization), we adopt a two-stage "distillation" strategy. First, we generate oracle labels by evaluating every expert on every sample and selecting the one with the minimum loss. Second, we train a separate Logistic Regression classifier (the Router) to predict this optimal expert given the input activation. During inference, for a Top-$K$ setting, we select the $K$ experts with the highest router probabilities.

### B.3.3. ROBUSTNESS EVALUATION PROTOCOL

To rigorously verify our claim that MoE architectures provide enhanced robustness against noise, we evaluate the models under two distinct noise injection scenarios:

1. **Gaussian Feature Noise:** This setting directly mirrors our theoretical assumption $\bar{X} = X + E$. We train the experts and router on clean data but evaluate them on test features corrupted by additive Gaussian noise $E \sim \mathcal{N}(0, \sigma^2 I)$. We vary the noise standard deviation $\sigma \in \{0.2, 0.5, 1.0, 2.0\}$.

2. **Input Perturbations:** To simulate real-world noise, we apply perturbations directly to the raw input text before feeding it into the T5 encoder. We consider *Word Swap* (randomly swapping adjacent words) and *Character Noise* (simulating keyboard errors).

**Metric: Performance Drop.** To isolate the robustness gain, we report the **Performance Drop**, defined as:

$$\text{Drop} = \text{Performance}_{\text{clean}} - \text{Performance}_{\text{noisy}} \tag{10}$$

where the score is Accuracy for balanced datasets (SST-2, MNLI, AG News) and Weighted F1 for the imbalanced dataset (CoLA). A **lower** drop indicates better robustness. Note that a negative drop implies the model performance improved slightly under noise, likely due to a regularization effect.

### B.3.4. EXPERIMENTAL RESULTS

We compare our MoE probing framework against a strong baseline: a global **Lasso** linear probe. For the MoE, we report the best result obtained by searching over regularization strengths.

The results are summarized in Tables 5, 6, and 7. Across almost all datasets and noise levels, **MoE configurations consistently exhibit a smaller performance drop than the Lasso baseline**. This advantage is particularly pronounced under high-intensity Gaussian noise (e.g., $\sigma = 1.0, 2.0$ in Table 5), where the dense Lasso baseline suffers significant degradation, while the sparse MoE maintains its efficacy. This strongly supports our theoretical insight that activation sparsity in MoEs acts as a filter, effectively suppressing feature noise.

### B.4. Experimental Details: Training MiniMind from Scratch

To empirically validate our theoretical findings regarding convergence speed and sample efficiency without the interference of pre-trained weights, we conducted a "train from scratch" experiment using the **MiniMind** framework (Gong, 2024). This setup allows for a controlled comparison between Dense and Mixture-of-Experts (MoE) architectures under strict iso-parameter conditions.

*Table 5.* We report the drop in metric (Accuracy or Weighted F1) when Gaussian noise $\mathcal{N}(0, \sigma^2)$ is added to the activations. Lower values indicate better robustness. The MoE models significantly outperform Lasso, especially at higher noise levels.

| Dataset | Noise Level | Lasso | MoE | | | |
|---|---|---|---|---|---|---|
| | | | $E=4, K=1$ | $E=4, K=2$ | $E=6, K=4$ | $E=8, K=6$ |
| **SST-2** | $\sigma = 0.2$ | 0.0115 | **-0.0046** | 0.0069 | 0.0023 | 0.0011 |
| | $\sigma = 0.5$ | 0.0493 | **0.0138** | 0.0206 | 0.0310 | **0.0138** |
| | $\sigma = 1.0$ | 0.0963 | **0.0241** | 0.0619 | 0.0711 | 0.0528 |
| | $\sigma = 2.0$ | 0.1686 | 0.0975 | 0.1101 | 0.1399 | **0.0952** |
| **CoLA** | $\sigma = 0.2$ | 0.0138 | **-0.0119** | 0.0056 | -0.0163 | -0.0034 |
| | $\sigma = 0.5$ | 0.0396 | 0.0013 | 0.0174 | **-0.0172** | -0.0067 |
| | $\sigma = 1.0$ | 0.0561 | 0.0444 | 0.0196 | **0.0048** | 0.0219 |
| | $\sigma = 2.0$ | 0.0847 | 0.0426 | 0.0637 | 0.0763 | **0.0310** |
| **MNLI** | $\sigma = 0.2$ | 0.0083 | 0.0037 | 0.0068 | 0.0016 | **-0.0004** |
| | $\sigma = 0.5$ | 0.0547 | 0.0410 | 0.0268 | 0.0222 | **0.0146** |
| | $\sigma = 1.0$ | 0.1458 | 0.1112 | 0.0963 | 0.0780 | **0.0645** |
| | $\sigma = 2.0$ | 0.2557 | 0.2120 | 0.2107 | 0.1898 | **0.1798** |
| **AG News** | $\sigma = 0.2$ | 0.0076 | 0.0078 | 0.0046 | **0.0022** | 0.0042 |
| | $\sigma = 0.5$ | 0.0549 | 0.0521 | 0.0307 | **0.0259** | 0.0293 |
| | $\sigma = 1.0$ | 0.1626 | 0.1413 | 0.1183 | 0.1121 | **0.0992** |
| | $\sigma = 2.0$ | 0.3166 | 0.2687 | 0.2658 | 0.2662 | **0.2580** |

**Model Architecture.** We utilized the MiniMind-MoE configuration as the backbone for both models.

- **Common Hyperparameters:** Both models share an identical configuration for the non-FFN components: 8 transformer layers, a hidden size ($d_{model}$) of 512, 8 attention heads, and a vocabulary size of approximately 6,400 (using a ByteLevel BPE tokenizer).

- **MoE Model:** The MoE variant employs a "Shared Expert + Routed Experts" architecture. It consists of $N = 4$ routed experts and $N_{shared} = 1$ shared expert. Each token selects $K = 2$ routed experts. To control the parameter count, the intermediate dimension of each expert FFN is set to $d_{inter}^{expert} = 1024$.

- **Dense Baseline:** To ensure a fair comparison based on total parameter count, the intermediate dimension of the dense FFN is set to $d_{inter}^{dense} = 5 \times 1024 = 5120$. This guarantees that the total parameters in the FFN layers of the dense model exactly match the sum of parameters of all experts in the MoE model.

It is important to note that while the total parameters are identical, the *active* parameters per token for the MoE model are significantly lower ($\approx 60\%$ of the dense model), as only the shared expert and top-2 routed experts are activated.

**Dataset and Training.** We used the `gongjy/minimind_dataset`, a high-quality pre-training corpus containing diverse general-purpose text.

- **Training Protocol:** Both models were trained from random initialization for 1 epoch (approx. 5000 steps) with a batch size of 32 using the AdamW optimizer ($lr = 5 \times 10^{-4}$).

- **Objectives:** The dense model was trained with standard Cross-Entropy loss. The MoE model was trained with Cross-Entropy loss plus a load-balancing auxiliary loss ($\alpha = 0.01$) to prevent routing collapse.

- **Evaluation:** We monitored the *Training Loss* to assess convergence speed and the *Validation Loss* on a held-out test set to evaluate sample complexity.

*Table 6.* The metric measures the performance drop after randomly swapping adjacent words (1 or 2 times). MoE models generally exhibit lower drops, indicating better resilience to structural perturbations.

| Dataset | Noise Level | Lasso | MoE | | | |
|---|---|---|---|---|---|---|
| | | | $E = 4, K = 1$ | $E = 4, K = 2$ | $E = 6, K = 4$ | $E = 8, K = 6$ |
| **SST-2** | Swap=1 | 0.0000 | **-0.0057** | 0.0034 | 0.0057 | 0.0046 |
| | Swap=2 | 0.0046 | -0.0023 | -0.0034 | -0.0023 | **-0.0057** |
| **CoLA** | Swap=1 | 0.1856 | 0.1103 | 0.1260 | 0.0902 | **0.0698** |
| | Swap=2 | 0.2338 | 0.1446 | 0.1764 | 0.1317 | **0.1073** |
| **MNLI** | Swap=1 | 0.0275 | 0.0193 | **0.0178** | 0.0194 | **0.0178** |
| | Swap=2 | 0.0502 | **0.0214** | 0.0277 | 0.0322 | 0.0291 |
| **AG News** | Swap=1 | -0.0005 | 0.0042 | 0.0020 | 0.0007 | **-0.0009** |
| | Swap=2 | 0.0039 | 0.0050 | **0.0011** | 0.0066 | 0.0075 |

*Table 7.* We simulate keyboard errors by replacing characters. MoE models maintain robustness, particularly on the CoLA dataset.

| Dataset | Noise Level | Lasso | MoE | | | |
|---|---|---|---|---|---|---|
| | | | $E = 4, K = 1$ | $E = 4, K = 2$ | $E = 6, K = 4$ | $E = 8, K = 6$ |
| **SST-2** | Char=1 | 0.0138 | 0.0011 | **-0.0034** | 0.0034 | -0.0080 |
| | Char=2 | 0.0252 | **-0.0138** | 0.0011 | -0.0011 | -0.0034 |
| **CoLA** | Char=1 | 0.0918 | 0.0535 | 0.0590 | 0.0620 | **0.0418** |
| | Char=2 | 0.1732 | 0.1187 | 0.1530 | 0.1322 | **0.1136** |
| **MNLI** | Char=1 | 0.0499 | 0.0517 | 0.0557 | 0.0511 | **0.0493** |
| | Char=2 | 0.0905 | 0.0941 | 0.0941 | 0.0930 | **0.0866** |
| **AG News** | Char=1 | 0.0046 | 0.0034 | **-0.0009** | 0.0011 | 0.0016 |
| | Char=2 | **0.0029** | 0.0089 | 0.0057 | 0.0054 | 0.0084 |

**Infrastructure.** The experiments were conducted on NVIDIA A100 GPUs using PyTorch. We implemented a custom training loop to support single-node multi-GPU parallel training, ensuring that both models were trained under identical system conditions.

### B.5. Experimental justification of the linear model assumption

To validate our theoretical insights in a more complex setting, we performed experiments on a synthetic dataset designed with a perfect modular structure. We trained both dense and MoE models on two tasks: a regression task with a single-layer linear network and a classification task with a two-layer non-linear network using ReLU activation. Both MoE models had access to a correctly routed input. We evaluated robustness by adding Gaussian noise $\mathcal{N}(0, \sigma^2)$ to the input features.

The results, shown in Table 8 and Table 9, demonstrate that MoE models are consistently more robust to noise than their dense counterparts across both linear and non-linear tasks. This supports our central claim that the robustness benefit is a fundamental property of the modular architecture.

*Table 8.* Robustness to noise (MSE). Lower MSE is better. The MoE model is more robust to noise, showing lower MSE and a smaller drop in performance.

| Noise Std ($\sigma$) | Dense w/ noise (MSE ↓) | MoE w/ noise (MSE ↓) | Dense-Robustness drop (↓) | MoE-Robustness drop (↓) |
|---|---|---|---|---|
| 0.1 | 3.13e-3 $\pm$ 4.45e-6 | 7.07e-4 $\pm$ 6.36e-6 | 2.10e-4 $\pm$ 2.18e-7 | -1.54e-5 $\pm$ 4.28e-6 |
| 0.2 | 9.88e-3 $\pm$ 6.85e-6 | 2.64e-3 $\pm$ 1.73e-5 | 3.41e-3 $\pm$ 3.45e-6 | 1.50e-5 $\pm$ 1.69e-5 |
| 0.3 | 1.57e-2 $\pm$ 1.17e-5 | 5.64e-3 $\pm$ 3.49e-5 | 1.41e-2 $\pm$ 1.49e-5 | 2.84e-4 $\pm$ 4.13e-5 |
| 0.4 | 1.88e-2 $\pm$ 2.44e-5 | 9.40e-3 $\pm$ 5.79e-5 | 3.39e-2 $\pm$ 2.47e-5 | 1.01e-3 $\pm$ 8.65e-5 |
| 0.5 | 2.03e-2 $\pm$ 5.37e-5 | 1.36e-2 $\pm$ 8.46e-5 | 6.15e-2 $\pm$ 4.07e-5 | 2.42e-3 $\pm$ 1.63e-4 |

*Table 9.* Robustness to noise (Accuracy) on a **2-layer non-linear classification task** ($\sigma = 0.1$). Higher accuracy is better. The MoE model achieves higher accuracy, and its performance drops less than the dense model's.

| Seed | Dense w/ noise (Acc. ↑) | MoE w/ noise (Acc. ↑) | Dense-Robustness drop (↓) | MoE-Robustness drop (↓) |
|---|---|---|---|---|
| 1 | 74.49% | 76.96% | 1.03% | 0.16% |
| 2 | 78.06% | 79.42% | 0.68% | 0.15% |
| 3 | 74.68% | 75.96% | 0.01% | -0.00% |
| 4 | 73.02% | 74.60% | 0.35% | 0.08% |

