# OpenReview forum: "Robustness of Mixtures of Experts to Feature Noise"
_ICML.cc/2026/Conference — ICML 2026 regular_

### Official Review · Reviewer_mjTt · 2026-03-11

**Soundness:** 3
**Presentation:** 3
**Significance:** 2
**Originality:** 3
**Overall Recommendation:** 5
**Confidence:** 3

**Summary:**

This paper asks why MoE models handle feature noise better than their dense counterparts. The authors formalize this in a linear regression setting where design matrices are block-diagonal, then compare a "dense" estimator (using the full model) against a "sparse" one (per-expert). They derive Bayes-optimal forms for both and demonstrate that the sparse estimator achieves lower generalization error - the intuition being that each expert's subspace isolates it from noise in other blocks. Beyond generalization, they also analyze perturbation robustness, convergence, and sample complexity. A QDA-based routing scheme is shown to be efficiently learnable. On the experimental side, there are synthetic validations plus linear probing experiments on frozen Llama-2 and Llama-3.1 across SST-2, CoLA, MNLI, and AG News under various noise regimes.

**Compliance With Llm Reviewing Policy:**

Affirmed.

**Key Questions For Authors:**

1. What happens to your results when the block-diagonal structure is only approximate? For instance, if each expert's features bleed slightly into neighboring blocks - does the sparse estimator still win, and can you characterize the conditions on cross-block magnitude where it breaks down?

2. In the MiniMind experiments, the MoE uses roughly 60% of active parameters per token. Is the comparison controlling for FLOPs per training step or total parameter count? That distinction is important for the efficiency claims.

3. Have you considered validating directly on a pre-trained MoE like Mixtral - comparing its noise robustness against a similarly-sized dense model? That would more directly test whether the theoretical predictions carry over.

**Limitations:**

yes

**Strengths And Weaknesses:**

Strengths

The core theory is well-executed. The closed-form comparison of generalization errors for both estimators, showing the sparse one dominates because each expert only contends with noise in its own subspace, is a clean result. The analysis also covers perturbation bounds and convergence behavior, which rounds out the theoretical picture. Using Bayes optimality as the proof strategy is a principled choice.

The framing around linear probing on frozen LLMs is a useful contribution. Treating MoE-based probes as routed linear predictors is a perspective I have not encountered before, and the visualization of block-diagonal activation patterns in Llama-2 post-TEAL-pruning provides good motivational evidence.

The paper is well-written. The progression from setup to theory to experiments is logical and the notation is consistent.

Weaknesses

My main concern is the distance between the theoretical model and actual MoE practice. Perfect block-diagonal modularity is a strong assumption; in real transformers, experts specialize partially with overlap. The authors acknowledge this, but I would have liked at least a brief theoretical exploration of approximate block structure - for example, what happens with small off-diagonal entries? Without that, the theory covers an idealized case that may not reflect how real MoE routing works.

The linear probing experiments are somewhat indirect as validation. They confirm that linear MoE-style estimators are more robust when applied to frozen representations, but that is a different question from whether end-to-end trained MoE transformers (Mixtral, for instance) exhibit the same noise-filtering advantage during training. The MiniMind experiments are closer to end-to-end, but with such a small model, it is difficult to draw strong conclusions. Even one experiment on a moderately-sized trained MoE would help.

The sample complexity analysis is less convincing than the earlier theory. Both estimators converge at O(n^{-2}) with the sparse one having a better constant, but this is framed as a "hypothesis" rather than proven, and the empirical curve-fitting does not clearly separate the two rates.

On baselines: Lasso alone is not sufficient. Ridge regression, Huber loss, or other robust regression approaches would help distinguish whether the benefit comes from modular sparsity specifically or from regularization more broadly.

Additionally, the mis-routing analysis involves expressions that are difficult to interpret. The accompanying remark provides some intuition, but the practical question remains unclear: under what realistic conditions does mis-routing hurt the MoE estimator more than the dense one?

---

> ### Author Rebuttal · Authors · 2026-03-31
>
> We thank the reviewer for the detailed and constructive feedback sincerely.
>
> 1. **Approximate block-diagonal structure:** We added a controlled overlap-sensitivity experiment (also described in our response to Reviewer 55Cw, Question 1).
> The results indicate that MoEs persist to be more robust over a broad overlap range, which supports the practical relevance of the mechanism beyond the exact block-diagonal case.
> 2. **Q2:** In the MiniMind experiment, parameter count rather than FLOPs are controlled. However, the active compute can be estimated. At the FFN level, the MoE uses approximately 60% of the dense FFN FLOPs since the router is a small linear layer whose cost is negligible relative to the FFN. The attention and other non-FFN components are unchanged between the two models. This corresponds roughly to 0.65–0.70 of the dense model’s FLOPs, depending on sequence length.
> 3. **Q3:** We appreciate the suggestion to validate our theoretical predictions on end-to-end trained architectures. We evaluated a Dense ViT-L/16 against a Sparse V-MoE-B/16 on the ImageNet-C Gaussian noise benchmark. To ensure a fair comparison, both models utilize state-of-the-art RandAugment training recipes—the "AugReg" suite for the dense baseline and the "strong" augmentation recipe for V-MoE. MoEs show clear robustness benefits.
>
>    Table 1: ImageNet-C Gaussian Noise Accuracy (%)
>
>    | Model | Clean | Sev 1 | Sev 3 | Sev 5 |
>    | :--- | :---: | :---: | :---: | :---: |
>    | Dense ViT-L (224px) | 84.33 | 80.65 | 74.19 | 46.57 |
>    | Sparse V-MoE (384px) | 84.82 | 82.35 | 77.91 | 59.65 |
>    | MoE Advantage ($\Delta$) | +0.49 | +1.70 | +3.72 | +13.08 |
>
>    **Structural Robustness:** While clean performance is nearly identical ($\Delta = 0.49\%$), the gap widens dramatically under noise. At the highest severity, the V-MoE achieves a 13.08% lead.
>
>    **Compute Efficiency:** While the V-MoE operates at a higher resolution (384px), it does so while utilizing only 37% of the active parameters of the dense model (114M vs. 307M). This demonstrates that sparse expert specialization provides a more robust representation than monolithic dense layers, even when accounting for differences in input scale.
>
>    These results confirm that the noise-filtering advantages of MoEs persist in state-of-the-art, end-to-end trained Transformers.
>
> 4. **Sample Complexity:** We appreciate this comment and agree that this part of the paper should be phrased more carefully. We do, however, respectfully think that Figure 4 already shows a consistent separation in excess risk between the dense and sparse estimators across the sample range. If this distinction was not sufficiently clear, that is likely due to our presentation rather than to the absence of a trend. In the revision, we will therefore improve the figure presentation.
>
> 5. **Baselines:** We thank the reviewer for the suggestion on stronger baselines. We agree that comparing only against a global Lasso probe is not sufficient to fully disentangle modular sparsity from generic regularization. To address this, we extended the probing experiments in two ways. First, we additionally evaluate on Qwen2.5-3B, providing a second modern LLM family. Second, we strengthen the dense baselines by including Ridge and Elastic Net in addition to Lasso. Across most settings, the MoE can achieve a better robustness performance.
>
>
>    Table 2: $\sigma=1.0$, Performace Drop $\downarrow$ (%)
>
>    | Dataset | Lasso | Ridge | Elastic Net | MoE |
>    | ------- | ----: | ----: | ----------: | --: |
>    | SST-2   |   5.96    |    6.53   |   **5.16**          |   5.39  |
>    | CoLA    |  6.91     |   9.55    |  6.97           |   **4.90**  |
>    | MNLI    |  8.47     |    9.22   |    5.02         |  **3.94**   |
>    | AG News |   2.92    |   3.49    |   **1.92**          | 3.27    |
>
>    Table 3: $\sigma=2.0$, Performace Drop $\downarrow$ (%)
>
>    | Dataset | Lasso | Ridge | Elastic Net | MoE |
>    | ------- | ----: | ----: | ----------: | --: |
>    | SST-2   |  10.78     |   12.27    |   10.55          |  **8.60**   |
>    | CoLA    |  10.25     |  12.39     |   12.19          |  **7.67**   |
>    | MNLI    | 11.61      |   10.45    |    9.31         |  **7.98**   |
>    | AG News |  5.19     |   5.98    |    **3.92**         |  7.75   |

---

> > ### Author Rebuttal · Reviewer_mjTt · 2026-04-04
> >
> > My concerns have been resolved, and I wish to retain my original score.

---

> > > ### Author Response · Authors · 2026-04-07
> > >
> > > We sincerely thank the reviewer again for the constructive comments and positive assessment of our work.

---

### Official Review · Reviewer_55Cw · 2026-03-12

**Soundness:** 4
**Presentation:** 4
**Significance:** 3
**Originality:** 3
**Overall Recommendation:** 5
**Confidence:** 4

**Summary:**

This paper investigates the theoretical underpinnings of why Mixture of Experts (MoE) models can outperform dense networks beyond the standard explanation of sheer parameter scaling. THe authors study an iso-parameter regime where inputs possess a latent block diagonal structure, but are corruptedby feature noise (Theorem 4.4). Through a simplified linear model framework, the paper demonstrates theoreticall that sparse expert activation filters noise from irrelevant feature blocks, leading to improved generalization, robustness to mis-routing, and faster convergence (verified empirically in Figure 3). The theoretical insights are corroborated by empirical results on synthetic data and real-world language tasks.

**Compliance With Llm Reviewing Policy:**

Affirmed.

**Final Justification:**

In summary, my original response captured my assessment of the paper. I appreciate the authors for their clarifications which allowed me to confidently maintain my score.

**Key Questions For Authors:**

* Generative Mismatch: The paper assume a block diagonal structure of the generative process which is defended in Figure 1 on large scale data. While Figure1 is compelling, it is clear that the generative process still does not exactly match this structure. Do you expect that these results extend when the generative process  does not exactly match the MoE architecture, or is it more accurate to think of the MoE architecture on this non-exact MoE generative process lying somewhere in between the dense and sparse generalization errors derived in Theorem 4.2.
* Joint Training vs Decoupled Routing: Theorem 4.2and its implications seem to represent a "best-case scenario" assuming routing is already solved (the z labels are known). Would you agree that Theorem 4.4 is arguably the more accurate theorem for characterizing realistic training methodologies where labels are estimated and noise exists? Subsequently, your findings seem to clash with the paper's primary narrative of robustness since "specialized experts excel when routing is correct but can be detrimental if routing fails significantly". Does this implicitly explain why jointly training the router and experts might be preferred over a decoupled approach? Comparing both methodologies empirically would greatly help justify your theoretical relaxation and provide further implications for your theoretical results.

**Limitations:**

The primary limitation in my view (which is addressed in the main text) is the relaxation of the MoE problem to the setting where the labels are estimated or known a priori. I would like to see an extended discussion on why this relaxation is appropriate and its implications for the theoretical results.

**Strengths And Weaknesses:**

Strengths:

* Strong Motivation & Relevance: The paper is highly relevant to current trends in scaling laws. As data becomes an increasingly critical component of scaling (e.g. Chinchilla), understanding how architectures efficiently filter noisy representations is vital. The problem is well-motivated and grounded in empirical observations of large-scale LLMs, particularly regarding activation sparsity. This strengthens the assumption that the ground truth generative process is an MoE/ follows a block diagonal structure.
* Clear Objectives & Implications: The objectives are clearly laid out in the introduction, and the implications of the work and every single result is clearly verified and articulated in the main text.
* Honest Theoretical Framing: While the theory is limited to the linear setting where the latent labels for the expert routing are known apriori, the authors are upfront about this and provide a compelling explanation for their methodological choice.
* Compelling Empirical Support: Figure 3 is particularly compelling in demonstrating the superior sample efficiency of the MoE model compared to the dense baseline.

Weaknesses:
* Clarity on the Frozen LLM Routing discussed in the introduction: The comparison regarding routing to frozen LLMs is interesting, but the setup is a little hard to parse. In particular, the concept of the "global linear baselines" and how it directly compares to the MoE-based probes could use clearer exposition in the main text.
* Transparency of Assumptions in the Abstract/Intro: The discussions in Section 3 and 4 implies that the ground truth routing to distinct experts is known a prior (or solved perfectly via some for mof clustering). While the authors provide convincing justification for this relaxation in Section 4.1, this is a significant departure from how traditional MoEs are jointly trained. This theoretical limitation should be emphasized earlier in the paper to set correct reader expectations (particularly in the abstract).
* Missing Definitions: Theorem 4.1 relies on a Quadratic Discriminant Analysis (QDA) based router, but this is not defined or explained anywhere in the main text. A brief formal definition before the theorem would improve readability and make the paper more self-contained.

---

> ### Author Rebuttal · Authors · 2026-03-31
>
> We sincerely thank the reviewer for the positive assessment and the thoughtful questions.
>
> 1. **Clarity on the Frozen LLM Routing:** In our paper, the main dense baseline is Lasso, i.e., a single global linear model trained on the full activation vector with feature selection induced by the $l_1$ penalty. The rationale is that it provides a natural dense comparison to ask whether modular routed sparsity offers benefits beyond simply selecting a sparse subset of features globally. Additionally, we have run stronger dense baselines such as Ridge and Elastic Net, as shown in Point 5 of our reply to Reviewer mjTt.
> 2. **Definition of QDA:** QDA (Quadratic Discriminant Analysis) is a generative classifier that models each class as a Gaussian distribution with class-specific covariance. Given a test point, it predicts the class with the largest discriminant score (equivalently, posterior under the Gaussian class model). Unlike LDA, QDA allows each class to have its own covariance matrix, which makes it well-suited to our setting where different experts/blocks may have different within-block covariance structures.
> 3. **Q1:** Figure 1 motivates approximate rather than exact block structure. Real data must not lie exactly in the idealized Theorem 4.2 regime. A more accurate interpretation is that realistic settings may lie between the dense and sparse idealized extremes, depending on how strong the cross-block interference is. The additional overlap experiment supports this interpretation: as off-block overlap increases, both estimators degrade, but the MoE continues to outperform the dense baseline over a broad range. This suggests that the proposed noise filtering mechanism is robust to moderate mismatch and degrades gradually rather than disappearing abruptly once perfect block structure is violated.
>
>    In the following experiment, we compare (i) a dense global linear classifier trained on all features, and (ii) a routed sparse estimator consisting of one linear classifier per expert trained only on its corresponding block. In this experiment, the router has access to the oracle to isolate the structural effect of approximate modularity from optimization chanllenges due to joint routing. In other words, this experiment is designed as a sensitivity analysis around the setting of Theorem 4.2, not as a simulation of full MoE training. We then introduce cross-block overlap by adding nuisance signal to all feature dimensions with tunable magnitud $\epsilon$ before adding Gaussian observation noise.
>
>    | Overlap ($\epsilon$) |      Dense Acc. | Sparse Acc. | Gap (Sparse - Dense) |
>    | ------------------ | --------------: | -----------------: | -------------------: |
>    | 0.00               | 0.6059 |    0.7025 |        0.0966       |
>    | 0.05               | 0.6063 |    0.7038 |              0.0974 |
>    | 0.10               | 0.6054 |    0.7036|              0.0982 |
>    | 0.20               | 0.6023 |    0.7021|              0.0998 |
>    | 0.30               | 0.5987 |    0.6991 |              0.1004 |
>    | 0.50               | 0.5921  |    0.6894|              0.0973 |
>    | 1.00               | 0.5659|    0.6571|              0.0912 |
>
> 4. **Q2:** Theorem 4.2 and Theorem 4.4 serve complemenatry roles. Theorem 4.2 isolates the structural best-case advantage of a sparse estimator under correct routing, while Theorem 4.4 characterizes how this advantage degrades when routing is imperfect. This separation is intentional, since our goal is first to identify the mechanism and then to analyze its sensitivity to routing errors, rather than to conflate both effects in a single jointly trained model.
>
> 5. **Joint vs. Decoupled Training:** We deliberately analyze the decoupled setting because our objective is to isolate the structural advantage of modular sparse computation under noisy features. Joint training couples two distinct phenomena: learning expert specialization and learning routing. Once these are entangled, it becomes much harder to identify which part is responsible for the robustness effect. Our decoupled analysis therefore serves as a controlled mechanism analysis. Moreover, this is not merely an artificial relaxation: recent theory suggests that expert feature learning can precede and guide router learning, meaning that expert learning can be the primary driver of the joint training dynamics [1]. From this perspective, analyzing the expert structure first is a theoretically meaningful path rather than an arbitrary simplification.
>
> [1] F. Liao and A. Kyrillidis, "Guided by the Experts: Provable Feature Learning Dynamic of Soft-Routed Mixture-of-Experts", AISTATS 2026.

---

> > ### Author Rebuttal · Reviewer_55Cw · 2026-04-02
> >
> > I thank the authors for addressing my questions and concerns. I will maintain my score.

---

> > > ### Author Response · Authors · 2026-04-07
> > >
> > > We sincerely thank the reviewer again for the constructive comments and positive assessment of our work.

---

### Official Review · Reviewer_ehDA · 2026-03-13

**Soundness:** 3
**Presentation:** 3
**Significance:** 3
**Originality:** 3
**Overall Recommendation:** 4
**Confidence:** 3

**Summary:**

In this paper, the author consider the parameter estimation problem for the linear models. The main objective is to highlight the advantage of spare MoE structure over dense estimation. The main contribution of this paper can be summarised as follows:

(R1) They demonstrate the benefit of spare MoE structure for gaining a much less error (Theorem 4.2) together with convergence rate for gradient descent algorithm (Theorem 4.6).

(R2) In particular, they study their problems based on the input-noise setting, which is common in modern AI model. Thus, they give an interesting inside about the robustness of their estimation.

(R3) They conduct several experiments based on several benchmarks, including SST2, CoLA, MNLI, etc, thus further corroborating their theoretical finding.

**Compliance With Llm Reviewing Policy:**

Affirmed.

**Key Questions For Authors:**

(Q1) Follow the (W1), would it be possible if we study the effect of non-linear activation?

(Q2) Another idea is to choose the noise randomly. I am wondering if we put each entry a Bernoulli random variable, if the value is 1, we put this entry with random noise, otherwise, we keep this value.

(Q3) In theorem 4.2, could you please explain how the selection of $p\_i$ can affect our model together with convergence rate?

**Limitations:**

yes

**Strengths And Weaknesses:**

- Strength:

(S1) This is the first work studying the effect of spare MoE and their advantage in parameter estimation in linear model,  This study gives us a very interesting insight about the MoE compared with other previous work.

(S2) The noise effect is well-formulated in this paper using a Gaussian noise, which makes this setting to be more practical.

(S3) The experiments are implemented in several LLM benchmarks, and strongly support theoretical findings.

(S4) The paper is well-written and easy to follow. No grammatical/typographic error is detected.

- Weakness:

(W1) Assumption about linear model does not fully capture modern neural network architecture. In fact, as we employ several non-linear activating function such as ReLU or softmax, the author may further investigate this issue in future research.

(W2) Despite the novelty of the findings, the result is somewhat incremental. For example, the linear model is universal in many researches, and people now tend to use more complicated setting (see W1). Also, calculation of Bayes risk and convergence velocity appears to follow a standard process.

---

> ### Author Rebuttal · Authors · 2026-03-31
>
> We thank the reviewer for recognizing the novelty of studying sparse MoE advantages in the noisy-feature setting. We especially appreciate the suggestions regarding nonlinear extensions and alternative noise distributions. Below, we address the concerns and the problems.
>
> 1. **Linear Model Assumption:** We agree that our formal theorems are stated for a linear setting. This is a deliberate modeling choice to isolate the structural effect of routed sparsity under feature noise in a tractable way. The point is not that modern MoEs are linear, but that the proposed noise-filtering mechanism is already visible at the linear readout level and is naturally instantiated in the linear probing setting on frozen representations. We will clarify this more explicitly. At the same time, the mechanism itself is not inherently restricted to linear models: whenever representations remain approximately modular, the same intuition extends to nonlinear experts, as confirmed by our empirical findings.
>
> 2. **Bernoulli distribution:** Our framework extends beyond Gaussian noise to random coordinate corruption schemes such as Bernoulli noise. If each coordinate is corrupted i.i.d., the perturbation can be written as a masked additive noise process; after centering, this yields a sub-Gaussian noise model with covariance depending on the corruption probability distribution. In that case, the same qualitative comparison persists. That means, dense models accumulate noise from many irrelevant coordinates, whereas sparse experts only see their routed subspace. We use Gaussian noise in the paper mainly because it leads to closed form Bayes estimators and especially transparent risk expressions. It is also a standard first approximation in high dimensional settings, where aggregate perturbations from many weak independent sources are often well-approximated by a Gaussian distribution.
>
>     We also tested a Bernoulli distribution noise model experimentally. The dense–sparse gap increases with the corruption probability, which is fully consistent with our mechanism.
>
>    | Signal std. | Corruption prob. $p$ |       Dense MSE |      Sparse MSE | Gap (Dense - Sparse) |
>    | ----------- | -------------------: | --------------: | --------------: | -------------------: |
>    | 0.7         |  0.05 | 0.9614  | 0.8930  |    0.0684 |
>    | 0.7         |   0.10 | 1.0464  | 0.9321  |  0.1143 |
>    | 0.7         |   0.20 | 1.1317  | 0.9052  |  0.2265 |
>    | 0.7         |  0.40 | 1.2601  | 0.9690 |  0.2911 |
>    | 1.5         | 0.05 | 4.6942  | 4.4544  | 0.2398 |
>    | 1.5         |  0.10 | 5.1037  | 4.6515  |  0.4522 |
>    | 1.5         |  0.20 | 5.4042  | 4.6623  |  0.7419 |
>    | 1.5 |  0.40 | 5.9131 | 4.6586  |  1.2545 |
>
> 4. **Effect of $p_i$:** In Theorem 4.2, $p_i$ represents the frequency with which block/expert i is selected. Its effect is intuitive: for the dense estimator, the effective signal covariance for block i is scaled to $p_i\Sigma_i$, so rare blocks in the dense estimator are statistically diluted and experience stronger shrinkage. By contrast, the sparse estimator (MoEs) conditions on the subspace of the signal and depends on $\Sigma_i$ rather than $p_i\Sigma_i$. Therefore, imbalance in the routing frequencies makes the dense estimator relatively worse, while the sparse estimator is much less affected. The same intuition also connects to optimization: smaller $p_i$ means fewer informative samples for some directions in the dense objective, which worsens conditioning and slows learning, whereas the routed decomposition alleviates this effect.

---

> > ### Author Rebuttal · Reviewer_ehDA · 2026-04-03
> >
> > Thank you for your comment. My concerns has been resolved, especially they provide an experiment for the case of Bernoulli distribution. The theoretical support for this Bernoulli distribution may be a further research direction. I decide to maintain the score.

---

> > > ### Author Response · Authors · 2026-04-07
> > >
> > > We sincerely thank the reviewer again for the constructive comments and positive assessment of our work.

---

### Official Review · Reviewer_TZmc · 2026-03-16

**Soundness:** 1
**Presentation:** 1
**Significance:** 2
**Originality:** 2
**Overall Recommendation:** 2
**Confidence:** 3

**Summary:**

The paper attempts to provide a theoretical framework for explaining why Mixture of Experts (MoE) models outperform dense models of the same parameter size. Using simplified linear models, the authors argue that the sparse activation inherent in MoEs acts as a natural filter against feature noise and claim that this sparsity grants MoEs superior generalization, faster training convergence, and better sample efficiency compared to dense counterparts.

**Compliance With Llm Reviewing Policy:**

Affirmed.

**Key Questions For Authors:**

The core theoretical framework assumes linear models, and the non-linear synthetic experiments rely on ReLU. Given that frontier models use highly non-linear soft activations (SwiGLU/GeGLU) that lack natural sparsity, is there any concrete mathematical evidence or large-scale empirical results that would prove that the proposed linear noise-filtering mechanism actually is present in non-toy, modern architectures?

**Limitations:**

Yes

**Strengths And Weaknesses:**

# Strengths

1. The paper provides a mathematical formalisation for the fact that MoEs converge faster and are more sample-efficient than dense models at equivalent parameter counts.

2. The authors attempt to mathematically map out why sparse routing might isolate feature noise, offering a theoretical lens for activation sparsity.

# Weaknesses

1. The paper relies on older references and reads like a recycled submission rather than a fresh contribution. The paper only cites 3 papers from 2025, which feels insufficient given how much research is conducted on MoEs and LLMs these days.

2. The main findings feel neither novel nor practical. MoE convergence have already been studied extensively in non-toy scenarios in several scaling law papers (e.g. https://arxiv.org/abs/2402.07871), and the fact that toy MoE models deal better with Gaussian noise in activations is merely an interesting curiosity which does not transfer to any meaningful applications, especially given very limited experimental scope.

3. The paper is very hard to read and feels inflated with theory that ultimately both lacks strong motivation and leads to very little insight.

---

> ### Author Rebuttal · Authors · 2026-03-31
>
> We thank the reviewer for the feedback. We appreciate the concerns regarding the freshness of the related work, the practical relevance of the proposed mechanism, and the applicability of our framework to modern LLMs. Below, we clarify the intended scope of the paper and address the questions.
>
> **1. References:** Thank you for this suggestion. We agree that the related-work section can be extended. In the revision, we will better position our work relative to recent MoE literature.
>
> Recent studies suggest that MoE gains are task-dependent, with stronger benefits for memorization than for reasoning under fixed active-parameter budgets [1]. Other recent theory analyzes expert specialization and forgetting in continual learning [2], convergence properties of MoE training through an EM / mirror-descent perspective [3], width-transfer and feature-learning guarantees via μ-parameterization for MoEs [4], expressivity gains from finer expert granularity [5] or from structured low-dimensional / sparse tasks [6], and loss-landscape geometry through linear mode connectivity in MoE architectures [7]. These directions are complementary to our paper: rather than focusing on task-dependent scaling, continual learning, optimization, loss-landscape geometry, or expressivity, we study a different theoretical question, namely robustness to feature noise under a parameter-matched comparison.
>
>    [1]Jelassi et al., *Mixture of Parrots: Experts improve memorization more than reasoning* (ICLR 2025);
>
>    [2]Li et al., *Theory on Mixture-of-Experts in Continual Learning* (ICLR 2025);
>
>    [3]Fruytier et al., *Learning Mixtures of Experts with EM: A Mirror Descent Perspective* (ICML 2025);
>
>    [4]Małaśnicki et al., *μ-Parameterization for Mixture of Experts* (arXiv 2025);
>
>    [5]Boix-Adsera and Rigollet, *The Power of Fine-Grained Experts: Granularity Boosts Expressivity in Mixture of Experts* (arXiv 2025);
>
>    [6]Wang and E, *On the Expressive Power of Mixture-of-Experts for Structured Complex Tasks* (NeurIPS 2025);
>
>    [7]Tran et al., *On Linear Mode Connectivity of Mixture-of-Experts Architectures* (NeurIPS 2025).
>
> **2. Relevance beyond the linear/ReLU setting.**
>
> Our goal is not to claim that the current theorems already constitute a full theory of jointly trained nonlinear MoEs. Rather, the paper isolates a specific mechanism: when representations exhibit latent modular structure and are corrupted by feature noise, MoE can improve robustness by restricting prediction to a subspace containing less noise.
>
> **(Practical) Relevance.** We provide four forms of evidence that this mechanism is relevant beyond toy ReLU models.
>
> First, the motivating block-structure observation in the introduction is obtained from LLaMA-family models rather than from a synthetic ReLU model, indicating that approximate modular structure is also visible in modern LLMs.
>
> Second, in our train from scratch Minimind experiment, which uses SwiGLU activation function, we compare dense and MoE models under strict iso-parameter control and observe improved training dynamics and sample-efficiency trends for MoE (Figures 2 and 3).
>
> Third, we further extend the linear probing study to Qwen2.5 3B and strengthen the dense baselines by adding Ridge and Elastic Net in addition to Lasso. You can find the details of this experiment in the point 5 of our reply to reviewer mjTt. This probing setting can be viewed as studying routed linear predictors on top of nonlinear learned representations, and therefore serves as a bridge between our tractable theory and modern architectures.
>
> Forth, we find a trained Sparse V-MoE-B/16 to be significantly more robust to feature noise than a Dense ViT-L/16  on ImageNet-C. Details can be found in Point 3 of our reply to Reviwer mjTt.
>
> **3. Motivation and insights:**
>  Our motivation is not to argue that MoEs outperform dense models simply because they have more total capacity. On the contrary, we hypothesize and prove that a MoE can outperform an iso-parameter dense model while activating only a small fraction of its parameters. If the underlying representation exhibits latent modularity—as suggested by the approximately block-diagonal activation patterns we observe in modern LLMs—then sparse expert activation acts as a noise filter: it suppresses irrelevant noisy feature blocks and preserves the signal-carrying subspace for each input. In a noiseless regime, a dense model could in principle recover the same structure; the advantage of MoEs appears precisely when realistic feature noise is present. This is a novel insight and mechanism that our theory isolates.

---

> > ### Author Rebuttal · Reviewer_TZmc · 2026-04-04
> >
> > Thanks for the response, but I think I will maintain my original score.
> >
> > First, the paper sells itself as study of the MoE properties, but mixes a lot of fundamentally different things together, e.g. Fig 1 presents TEAL - a MoEfied **dense** model, which is optimised in an entirely different way than standard sparse MoE trained in practice.
> >
> > Second, the theory lacks convincing empirical verification on practical, end-to-end architectures, relying primarily on toy models and linear probing.
> >
> > Finally, severe formatting issues, such as unreadable, cramped tables and equations bleeding into the margins, fall well below the presentation standards expected at ICML.
> >
> > All in all, in my understanding the paper would require a major revision to be publication worthy at a conference such as ICML. While I might partially see how other reviewers might light the theoretical side of the paper, I still believe that the messy motivation and bad presentation quality warrant rejection.

---

> > > ### Author Response · Authors · 2026-04-07
> > >
> > > We thank the reviewer for their reply.
> > >
> > > 1. TEAL is included only as a motivating visualization to show that approximate block structure can be observed in real LLM representations. It is not part of the formal theory, and none of our main results rely on TEAL. We can also observe such block structure without TEAL. In other words, TEAL is used only to motivate the realism of the modularity assumption, not to establish the main claim.
> > >
> > > 2. In addition to our primary theoretical contribution, we have provided strong empirical support of our main claims in real world experimental settings. In Point 2 of our rebuttal, we provided an end-to-end experiment on a real vision benchmark, showing that a trained Sparse V-MoE-B/16 is substantially more robust to feature corruption than a Dense ViT-L/16 on ImageNet-C. Together with the Minimind train-from-scratch experiment already included in the paper, as well as the extended probing results on additional modern LLM backbones, we have demonstrated that the proposed mechanism is not confined to theoretically tractable settings. We therefore kindly request the reviewer to take the empirical verification of our theory on practical, end-to-end architectures into account in their assessment.
> > >
> > > 3. Certainly, we will respect ICML formatting requirements and apologize if there exist any problems.

---

### Decision · Program_Chairs · 2026-04-30

**Decision:**

Accept (regular)

**Comment:**

This paper studies whether the advantage of Mixture-of-Experts models can arise not only from scale, but from structure. In a parameter-matched setting, it argues that when inputs have latent modular structure and are corrupted by feature noise, sparse expert activation can act as a noise filter, potentially yielding lower generalization error, greater robustness to perturbations and faster optimization than a dense estimator in the paper’s stylized setting. The reviewers viewed the paper as timely and theoretically motivated, with a clear central mechanism and encouraging synthetic, probing and smaller end-to-end experiments. At the same time, they felt the work is stronger as an idealized mechanism study than as a full explanation of practical modern MoEs: the main concerns were how far the linear, block-structured, partially decoupled-routing analysis carries to jointly trained nonlinear architectures, along with requests for stronger practical baselines, broader empirical validation and, in one review, substantial presentation cleanup. Overall, the discussion points to a positive but qualified assessment: an elegant and interesting theoretical story, with promising but still limited evidence for its reach beyond the stylized setting.